# Explicit Flow Matching: On The Theory of Flow Matching Algorithms with Applications

## Abstract

This paper proposes a novel method, Explicit Flow Matching (ExFM), for training and analyzing flow-based generative models. ExFM leverages a theoretically grounded loss function, ExFM loss (a tractable form of Flow Matching (FM) loss), to demonstrably reduce variance during training, leading to faster convergence and more stable learning. Based on theoretical analysis of these formulas, we derived exact expressions for the vector field (and score in stochastic cases) for model examples (in particular, for separating multiple exponents), and in some simple cases, exact solutions for trajectories. In addition, we also investigated simple cases of diffusion generative models by adding a stochastic term and obtained an explicit form of the expression for score. While the paper emphasizes the theoretical underpinnings of ExFM, it also showcases its effectiveness through numerical experiments on various datasets, including high-dimensional ones. Compared to traditional FM methods, ExFM achieves superior performance in terms of both learning speed and final outcomes.

## 1 Introduction

In recent years, there has been a remarkable surge in Deep Learning, wherein the advancements have transitioned from purely neural networks to tackling differential equations. Notably, Diffusion Models [16] have emerged as key players in this field. This models transform a simple initial distribution, usually a standard Gaussian distribution, into a target distribution via a solution of Stochastic Differentiable Equation (SDE) [1] or Ordinary Differentiable Equation (ODE)[2] with right-hand side representing a trained neural network. The Conditional Flow Matching (CFM) [9] technique, which we focus on in our research, is a promising approach for constructing probability distributions using conditional probability paths, which is notably a robust and stable alternative for training Diffusion Models. The development of the CFM-based approach includes various techniques and heuristics [4, 7, 13] aimed at improving convergence or quality of learning or inference. For example, in the works [19, 20, 10] it was proposed to straighten the trajectories between points by different methods, which led to serious modifications of the learning process. We refer the reader for, example, to the paper [20] where different FM-based approaches are summarised, and to the paper [9] for the connection between Diffusion Models and CFM.

In our work, we introduced an approach which we called Explicit Flow Matching (ExFM), to consider the Flow Matching framework theoretically by modifying the loss and writing the explicit value of the vector field. Strictly speaking, the presented loss is a tractable form of the FM loss, see Eq. (5) of [9]. Base on this methods we can improve the convergence of the method in practical examples reducing the variance of the loss, but the main focus of our paper is on theoretical derivations.

Our method allows us to write an expression for the vector field in closed form for quite simple cases (Gaussian distributions), however, we note that Diffusion Models framework in the case of a Gaussian Mixture of two Gaussian as a target distribution is still under investigation, see recent publications [15, 8].

Our main contributions are:

1. A tractable form of the FM loss is presented, which reaches a minimum on the same function as the loss used in Conditional Flow Matching, but has a smaller variance;

2. The explicit expression in integral form for the vector field delivering the minimum to this loss (therefore for Flow Matching loss) is presented.

3. As a consequence, we derive expressions for the flow matching vector field and score in several particular cases (when linear conditional mapping is used, normal distribution, etc.);

4. Analytical analysis of SGD convergence showed that our formula have better training variance on several cases;

5. Numerical experiments show that we can achieve better learning results in fewer steps.

## 1.1 Preliminaries

Flow matching is well known method for finding a flow to connect samples from two distribution with densities $\rho_0$ and $\rho_1$. It is done by solving continuity equation with respect to the time dependent vector field $\overline{v}(x, t)$ and time-dependent density $\rho(x, t)$ with boundary conditions:

$$\begin{cases} \dfrac{\partial \rho(x,t)}{\partial t} = -\operatorname{div}(\rho(x,t)\overline{v}(x,t)), \\ \rho(x,0) = \rho_0(x), \quad \rho(x,1) = \rho_1(x). \end{cases} \tag{1}$$

Function $\rho(x, t)$ is called *probability density path*. Typically, the distribution $\rho_0$ is known and it is chosen for convenience reasons, for example, as standard normal distribution $\rho(x) = \mathcal{N}(x \mid 0, I)$. The distribution $\rho_1$ is unknown and we only know the set of samples from it, so the problem is to approximate the vector field $v(x, t) \approx \overline{v}(x, t)$ using these samples. To make problem (1) well defined, one usually imposes additional regularity conditions on the densities, such as smoothness. The rigorous justification of the obtained results we put in the Appendix, leaving the general formulations of theorems and ideas in the main text.

From a given vector field, we can construct a *flow* $\phi_t$, *i. e.*, a time-dependent map, satisfying the ODE $\frac{\partial \phi_t(x)}{\partial t} = v(\phi_t(x), t)$ with initial condition $\phi_0(x) = x$. Thus, one can sample a point $x_0$ from the distribution $\rho_0$ and then using this ODE obtain a point $x_1 = \phi_1(x_0)$ which have a distribution approximately equal to $\rho_1$. For given boundary $\rho_0$ and $\rho_1$, the vector field or path solutions are not the only solutions, but if we have found any solution, it will already allow us to sample from the unknown density $rho_1$. However, if the problem is more narrowly defined, *e. g.*, one needs to have a map that is close to the Optimal Transport (OT) map, we have to impose additional constraints.

The problem of finding any vector field $v$ is solved in conditional manner in the paper [9], where so-called Conditional Flow Matching (CFM) is present. Namely, the following loss function was introduced for the training a model $v_\theta$ which depends on parameters $\theta$

$$L_{\text{CFM}}(\theta) = \mathbb{E}_t \mathbb{E}_{x_1, x_0} \big\| v_\theta(\phi_{t, x_1}(x_0),\, t) - \phi'_{t, x_1}(x_0) \big\|^2, \tag{2}$$

where $\phi_{t,x_1}(x_0)$ is some flow, conditioned on $x_1$ (one can take $\phi_{t,x_1}(x_0) = (1 - t)x_0 + tx_1 + \sigma_s t x_0$ in the simplest case, where $\sigma_s > 0$ is a small parameter need for this map to be invertable at any $0 \leq t \leq 1$). Hereinafter the dash indicates the time derivative. Time variable $t$ is uniformly distributed: $t \sim \mathcal{U}[0, 1]$ and random variables $x_0$ and $x_1$ are distributed according to the initial and final distributions, respectively: $x_0 \sim \rho_0$, $x_1 \sim \rho_1$. Below we omit specifying of the symbol $\mathbb{E}$ the distribution by which the expectation is taken where it does not lead to ambiguity.

## 1.2 Why new method?

Model training using loss (2) have the following disadvantage: during training, due to the randomness of $x_0$ and $x_1$, significantly different values can be presented for model as output value at close model

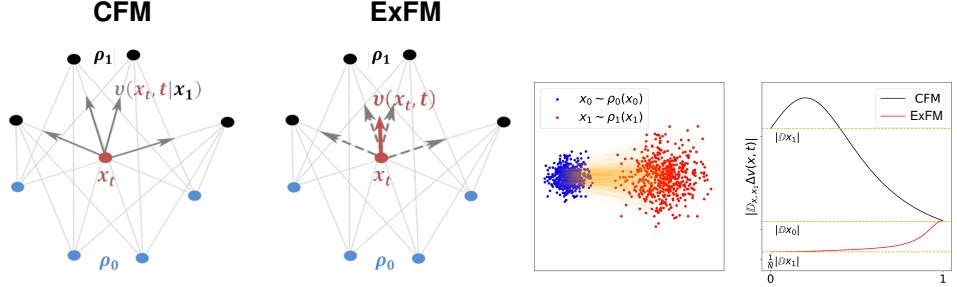

Figure 1: (Left) The key novelty of our approach is that in classical CFM, highly divergent directions can appear in a small spatial area at similar times (left part). In our approach (right part) we average over these vectors, training the model on a smoothed unnoised vector field. (Right) The comparison evaluated dispersion norm over time parameter $t$ for CFM and ExFM in matching standard Gaussian $\rho_0 = \mathcal{N}(0, I)$ to general Gaussian $\rho_1 = \mathcal{N}(\mu, \sigma^2 I)$ distributions. The y-axis represents the sum of dispersion vector components, denoted as $|\mathbb{D}_{x,x_1}\Delta v(x,t)|$. The left panel illustrates samples drawn from the $\rho_0$ and $\rho_1$ distributions, as well as the corresponding flows. The right panel depicts the dispersion trend over time for both CFM (black line) and ExFM (red line) objectives. The dotted lines correspond to the dispersion levels (in top-down order $|\mathbb{D}x_1|$, $|\mathbb{D}x_0|$, $|\mathbb{D}x_1|/N$.

79 argument values $(x_t, t)$. Indeed, a fixed point $x_t = \phi_{t,x_1}(x_0)$ can be obtained by an infinite set of $x_0$
80 and $x_1$ pairs, some of which are directly opposite, and at least for small times $t$ the probability of these
81 different directions may not be significantly different. At the same time, data $\phi'_{t,x_1}(x_0)$ on which the
82 model learns significantly different for such different positions of pairs $x_0$ and $x_1$. Thus, the model is
83 forced to do two functions during training: generalize and take the mathematical expectation (clean
84 the data from noise).

85 In our approach, see Fig. 1(a), we feed the model input with cleaned data with small variance. Thus,
86 the model only needs to generalize the data, which happens much faster (in fewer training steps).

87 Moreover, in the process of constructing the modified loss, we have developed the exact formula for
88 the vector field, see Eq. (11), (34). The existence of an explicit formula for the vector field is of great
89 importance not only from a theoretical but also from a practical point of view.

## 2 Main idea

### 2.1 Modified objective

92 Lets expand the last two mathematical expectations in the loss (2) and substitute variables using
93 map $\phi_{t,x_1}$, passing from the point $x_0$ to its position $x_t = \phi_{t,x_1}(x_0)$ at time $t$:

$$\mathbb{E}_{x_1,x_0}\big\|v_\theta(\phi_{t,x_1}(x_0),\, t) - \phi'_{t,x_1}(x_0)\big\|^2 = \iint \big\|v_\theta(\phi_{t,x_1}(x_0),\, t) - \phi'_{t,x_1}(x_0)\big\|^2 \rho_0(x_0)\rho_1(x_1)\mathrm{d}x_0\mathrm{d}x_1$$

$$= \iint \big\|v_\theta(x_t,\, t) - \phi'_{t,x_1}\big(\phi^{-1}_{t,x_1}(x_t)\big)\big\|^2 \underbrace{\det\left[\left.\partial\phi^{-1}_{t,x_1}(x)/\partial x\right|_{x=x_t}\right] \rho_0\big(\phi^{-1}_{t,x_1}(x_t)\big)}_{\rho_{x_1}(x_t,t)} \rho_1(x_1)\,\mathrm{d}x_t\,\mathrm{d}x_1$$

$$= \mathbb{E}_{x_1,x_t\sim\rho_{x_1}(\cdot,t)}\big\|v_\theta(x_t,\, t) - \phi'_{t,x_1}\big(\phi^{-1}_{t,x_1}(x_t)\big)\big\|^2. \quad (3)$$

94 We assume, that the map $\phi_{t,x_1}$ is invertible at each $0 < t < 1$, *i.e.* that $\phi^{-1}_{t,x_1}(x_t)$ exits on this
95 time interval and for all $x_t = \{\phi_t(x_0) \mid \forall x_0 : \rho(x_0) > 0\}$. Eq. (3) can be seen as a transition
96 from expectation on the variable $x_0 \sim \rho_0$ to expectation on the variable $x_t \sim \rho_{x_1}(\cdot, t)$, where
97 $\rho_{x_1}(x,t) = [\phi_{t,x_1}]_*\rho_0(x) := \rho_0\big(\phi^{-1}_{t,x_1}(x)\big)\det\big[\partial\phi^{-1}_{t,x_1}(x)/\partial x\big]$. See paper [5] for details about the
98 push-forward operator "*". Our representation (3) is very similar to expression (9) of the cited
99 paper [9], only we write it in terms of the conditional flow rather than the conditional vector field.

To obtain the modified loss, we return to end of the standard CFM loss representation in (3). It is written as the expectation over two random variables $x_1$ and $x_t$ having a common distribution density

$$\{x_1, x_t\} \sim \rho_j(x_1, x_t, t) = \rho_{x_1}(x_t, t)\rho_1(x_1), \tag{4}$$

which, generally speaking, is not factorizable. Let us rewrite this expectations in terms of two independent random variables, each of which have its marginal distribution. The marginal distribution $\rho_m$ of $x_t$ can be obtained via integration:

$$\rho_m(x_t, t) = \int \rho_j(x_1, x_t, t)\, \mathrm{d}x_1 = \int \rho_{x_1}(x_t, t)\rho_1(x_1)\, \mathrm{d}x_1\,, \tag{5}$$

while the marginal distribution of $x_1$ is just (unknown) function $\rho_1$. Let for convenience $w(t, x_1, x) = \phi'_{t,x_1}\big(\phi_{t,x_1}^{-1}(x)\big)$[1]. We have

$$L_{\mathrm{CFM}}(\theta) = \mathbb{E}_{t,x_1,x_t \sim \rho_{x_1}(\cdot,t)}\|v_\theta(x_t,\, t) - w(t, x_1, x_t)\|^2 =$$

$$\int_0^1 \iint \|v_\theta(x_t,\, t) - w(t, x_1, x_t)\|^2 \rho_{x_1}(x, t)\rho_1(x_1)\, \mathrm{d}x_t\, \mathrm{d}x_1 \mathrm{d}t =$$

$$\int_0^1 \iint \|v_\theta(x_t,\, t) - w(t, x_1, x_t)\|^2\, \left(\rho_{x_1}(x_t,t)/\rho_m(x_t,t)\right) \rho_m(x_t, t)\rho_1(x_1)\, \mathrm{d}x_t\, \mathrm{d}x_1 \mathrm{d}t =$$

$$\mathbb{E}_{t,x_1,x \sim \rho_m(\cdot,t)}\|v_\theta(x,\, t) - w(t, x_1, x)\|^2\, \rho_c(x|x_1, t)/\rho_1(x_1), \tag{6}$$

where we introduce a conditional distribution

$$\rho_c(x|x_1, t) := \rho_{x_1}(x, t)\rho_1(x_1)/\rho_m(x, t) := \rho_{x_1}(x, t)\rho_1(x_1) \Big/ \int \rho_{x_1}(x, t)\rho_1(x_1)\, \mathrm{d}x_1. \tag{7}$$

The key feature of the representation (6) is that the integration variables $x_1$ and $x$ are independent. Thus, we can evaluate them using Monte Carlo-like schemes in different ways. However, we go further and make a modification to this loss to reduce the variance of Monte Carlo methods.

## 2.2 New loss and exact expression for vector field

Note that so far the expression for $L_{\mathrm{CFM}}$ have not changed, it has just been rewritten in different forms. Now we change this expression so that its numerical value, generally speaking, may be different, but the derivative of the model parameters will be the same. We introduce the following loss

$$L_{\mathrm{ExFM}}(\theta) = \mathbb{E}_t \mathbb{E}_{x \sim \rho_m}\left\|v_\theta(x,\, t) - \mathbb{E}_{x_1 \sim \rho_1} w(t, x_1, x)\rho_c(x|x_1, t)/\rho_1(x_1)\right\|^2 =$$

$$\int_0^1 \int \left\|v_\theta(x,\, t) - \int w(t, x_1, x) \times \rho_c(x|x_1, t)\, \mathrm{d}x_1\right\|^2 \rho_m(x, t)\, \mathrm{d}x\, \mathrm{d}t. \tag{8}$$

**Theorem 2.1.** *Losses $L_{CFM}$ in Eq. (2) and $L_{ExFM}$ in Eq. (8) have the same derivative with respect to model parameters:*

$$\mathrm{d}L_{CFM}(\theta)/\mathrm{d}\theta = \mathrm{d}L_{ExFM}(\theta)/\mathrm{d}\theta\,. \tag{9}$$

Proof is in the Appendix A.1.

In the presented loss $L_{\mathrm{ExFM}}$, the integration (outside the norm operator) proceeds on those variables on which the model depends, while inside this operator there are no other free variables. Thus, using this kind of loss, it is possible to find an exact analytical expression for the vector field for which the minimum of this loss is zero (unlike the loss $L_{\mathrm{CFM}}$). Namely, we have

$$v(x, t) = \int w(t, x_1, x)\rho_c(x|x_1, t)\, \mathrm{d}x_1\,. \tag{10}$$

We can obtain the exact form of this vector field given the particular map $\phi_{t,x_1}$. For example, the following statement holds:

---

[1]Note, that $w(t, x_1, x)$ is the conditional velocity at the given point $x$.

**Corollary 2.2.** *Consider the linear conditioned flow $\phi_{t,x_1}(x_0) = (1-t)x_0 + tx_1$ which is inevitable as $0 \le t < 1$. Then $w(t, x_1, x) = \frac{x_1 - x}{1-t}$, $\rho_{x_1}(x, t) = \rho_0\left(\frac{x - x_1 t}{1-t}\right)\frac{1}{(1-t)^d}$ and the loss $L_{ExFM}$ in Eq. (8) reaches zero value when the model of the vector field have the following analytical form*

$$v(x,t) = \int (x_1 - x)\rho_0\left(\frac{x - x_1 t}{1-t}\right)\rho_1(x_1)\,\mathrm{d}x_1 \Big/ \left((1-t)\int \rho_0\left(\frac{x - x_1 t}{1-t}\right)\rho_1(x_1)\,\mathrm{d}x_1\right). \quad (11)$$

*This is the exact value of the vector field whose flow translates the given distribution $\rho_0$ to $\rho_1$.*

Complete proofs are in the Appendix A.3.1. Note that the result (11) is not totally new, for example, a similar result (though in the form of a general expression rather than an explicit formula), was given in [19], Eq. (9). However, our contribution consists of both the general form (10) and practical and theoretical conclusions from it (see below).

*Remark* 2.3. In the case of the initial and final times $t = 0,\ 1$, Eq. (11) is noticeably simpler

$$v(x,0) = \mathbb{E}_{x_1} x_1 - x = \int x_1 \rho_1(x_1)\,\mathrm{d}x_1 - x. \quad v(x,1) = x - \int x_0 \rho_0(x_0)\,\mathrm{d}x_0. \quad (12)$$

This expression for the initial velocity means that each point first tends to the center of mass of the unknown distribution $\rho_1$ regardless of its initial position.

**Extensions to SDE** Now let the conditional map be stochastic: $\phi_{t,x_1} = (1-t)x_0 + tx_1 + \sigma_e(t)\epsilon$, where $\epsilon \sim \mathcal{N}(0,1)$. Typically, $\sigma_e(0) = \sigma_e(1) = 0$, for example, $\sigma_e(t) = t(1-t)\sigma_e$.

Note that this formulation covers (with appropriate selection of the $\sigma_e(t)$ parameter) the case of diffusion models [20].

Then, we can write the exact solution for a so-called *score and flow matching* objective (see [20] for details)

$$\mathcal{L}_{[\mathrm{SF}]^2\mathrm{M}}(\theta) = \mathbb{E}\Big[\underbrace{\|v_\theta(x,t) - u_t^\circ(x)\|^2}_{\text{flow matching loss}} + \lambda(t)^2 \underbrace{\|s_\theta(x,t) - \nabla \log p_t(x)\|^2}_{\text{score matching loss}}\Big].$$

that corresponds to this map. In the last expression, the following explicit conditional expressions are considered in the cited paper for the case $\sigma_e(t) = \sqrt{t(1-t)}\sigma_e$

$$u_t^\circ(x) = \frac{1-2t}{t(1-t)}(x - (tx_1 + (1-t)x_0)) + (x_1 - x_0), \quad \nabla \log p_t(x) = \frac{tx_1 + (1-t)x_0 - x}{\sigma_e^2 t(1-t)}.$$

The exact solution (our result, explicit analog of the Eq. (10) from [20]) under consideration has the form (44) and (46) and, for example for the for the Gaussian $\rho_0$ this expressions reduced to the Eq. (49) and (50), correspondingly. See Appendix E for the details on this case.

**Simple examples** Consider the case of Standard Normal Distribution as $\rho_0$ and Gaussian Mixture of two Gaussians as $\rho_1$. Vector field have a closed form (37) in this case, and we can fast numerically solve ODE for trajectories. Random generated trajectories and plot of the vector field are shown on Fig. 2 (a)–(b). Detailed explanation of this case is in the Sec. D.2. Another example is related to the case of a stochastic map in the form of Brownian Bridge, which briefly described in the last paragraph and considered in Sec. E.3.2 in details, see Fig. 2 (c)–(f). Note that at some $\sigma_e$ values the trajectories are a little bit straightened in this case compared to the usual linear map, if we compare cases on the Fig. 6.

## 2.3 Training scheme based on the modified loss

Let us consider the difference between our new scheme based on loss $L_{\mathrm{ExFM}}$ and the classical CFM learning scheme. As a basis for the implementation of the learning scheme, we take the open-source code[2] from the works [20, 19].

Consider a general framework of numerical schemes in classical CFM. We first sample $m$ random time variables $t \sim \mathcal{U}[0,1]$. Then we sample several values of $x$. To do this, we sample a certain number $n$ samples $\{x_0^i\}_{i=1}^n$ from the "noisy" distribution $\rho_0$, and the same number $n$ of samples $\{x_1^i\}_{i=1}^n$ from

---

[2] https://github.com/atong01/conditional-flow-matching

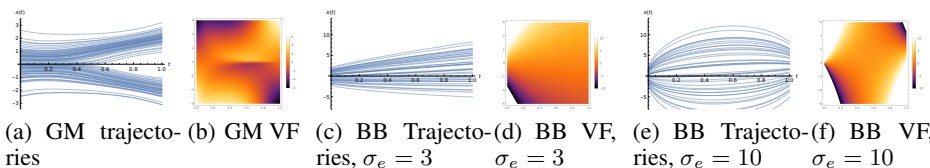

| (a) GM trajectories | (b) GM VF | (c) BB Trajectories, $\sigma_e = 3$ | (d) BB VF, $\sigma_e = 3$ | (e) BB Trajectories, $\sigma_e = 10$ | (f) BB VF, $\sigma_e = 10$ |

Figure 2: Trajectories and vector field obtained in simple cases: (a) $N = 80$ random trajectories from $\mathcal{N}\left(\cdot \,\middle|\, 0, 1^2\right)$ to GM; (b) 2D plot of the vector field in this case (c)–(f) $N = 40$ random trajectories from $\mathcal{N}\left(\cdot \,\middle|\, 0, 1^2\right)$ to $\mathcal{N}\left(\cdot \,\middle|\, 2, 3^2\right)$ and 2D plot of the vector fieldfor different $\sigma_e$ for the Brownian Bridge map

the unknown distribution $\rho_1$. Then we pair them (according to some scheme), and get $n$ samples as $x^{j,i} = \phi_{t^j, x_1^i}(x_0^i)$ (e. g. a linear combination in the simple case of linear map: $x^{j,i} = (1-t^j)x_0^i + t^j x_1^i$), $\forall i = 1, 2, \ldots, n;\ \forall j = 1, 2, \ldots, m$. Note, than one of the variable $n$ or $m$ (or both) can be equal to 1.

At the step 2, the following discrete loss is constructed from the obtained samples

$$L_{\text{CFM}}^d(\theta) = \sum_{j=1}^{m} \sum_{i=1}^{n} \left\| v_\theta(x^{j,i}, t^j) - \phi'_{t^j, x_1^i}(x_0^i) \right\|^2. \tag{13}$$

Finally, we do a standard gradient descent step to update model parameters $\theta$ using this loss.

The first and last step in our algorithm is the same as in the standard algorithm, but the second step is significantly different. Namely, we additionally generate a sufficiently large number $N \gg n \cdot m$ of samples $\overline{x}_1$ from the unknown distribution $\rho_1$, sampling $(N - n)$ new samples and adding to it the samples $\{x_1^i\}_1^n$ that are already obtained on the previous step.

Then we form the following discrete loss which replaces the integral on $x_1$ in $L_{\text{ExFM}}$ by its evaluation $v^d$ by self-normalized importance sampling or rejection sampling (see Appendix B for details)

$$L_{\text{ExFM}}^d(\theta) = \sum_{j=1}^{m} \sum_{i=1}^{n} \left\| v_\theta(x^{j,i}, t^j) - v^d(x^{j,i}, t^j) \right\|^2. \tag{14}$$

For example, if we use self-normalized importance sampling and assume that the Jacobian $\det\left[\partial \phi_{t,x_1}^{-1}(x)/\partial x\right]$ do not depend on $x_1$, we can write

$$v^d(x, t) = \left( \sum_{k=1}^{N} w(t, \overline{x}_1^k, x) \rho_0\left(\phi_{t, \overline{x}_1^k}^{-1}(x)\right) \right) \Bigg/ \sum_{k=1}^{N} \rho_0\left(\phi_{t, \overline{x}_1^k}^{-1}(x)\right). \tag{15}$$

**Theorem 2.4.** *Under mild conditions, the error variance of the integral gradient* (9) *using the Monte Carlo method* (14) *is lower than using formula* (13) *with the same number $n \cdot m$ of samples for $\{x\}$.*

Sketch of the proof is in the Appendix A.2. The steps of our scheme are formally summarized in Algorithm 1.

**Particular case of linear map and Gaussian noise**  Let $\phi_{t,x_1}$ be the linear flow: $\phi_{t,x_1}(x_0) = (1 - t)x_0 + tx_1$. and consider the case of standard normal distribution for the initial density $\rho_0$: $\rho_0(x) \sim \mathcal{N}(x \mid 0, I)$. Then in the case of using self-normalized importance sampling, we have

$$v^d(x, t) = \sum_{k=1}^{N} \frac{\overline{x}_1^k - x}{1 - t} \left(\text{SoftMax}(Y^1, \ldots, Y^N)\right)_k, \quad \text{where} \quad Y^k = -\frac{1}{2} \frac{\left\| x - t \cdot \overline{x}_1^k \right\|_{\mathbb{R}^d}^2}{1 - t}. \tag{16}$$

Here, the lower index $k$ in SoftMax stands for the $k$-th component, and the SoftMax operation itself came about due to exponents in the Gaussian density as a more stable substitute for computing than directly through exponents.

**Extension of other maps and initial densities $\rho_0$**  Common expression (10) can be reduced to closed form for the particular choices of density $\rho_0$ and map $\phi$ (consequently, expression for $w$). We summarise several known approaches for which FM-based techniques can be applied in Table 1[3]. See Appendix C and D for derivations of formulas and for more extensions.

---

[3]The idea and common structure of the Table is taken from [20]

Table 1: Correspondence between some methods which can reduced to FM framework and our theoretical descriptions of them.

| Probability Path | $q(z)$ | $\mu_t(z)$ | $\sigma_t$ | Explicit expressions: vector field (VF) and score (S) |
|---|---|---|---|---|
| Var. Exploding [17] | $\rho_1(x_1)$ | $x_1$ | $\sigma_{1-t}$ | VF: (32) |
| Var. Preserving [6] | $\rho_1(x_1)$ | $\alpha_{1-t}x_1$ | $\sqrt{1-\alpha_{1-t}^2}$ | VF: (31) |
| Flow Matching [9] | $\rho_1(x_1)$ | $tx_1$ | $t\sigma_s - t + 1$ | VF: (11) if $\sigma = 0$; and (26) |
| Independent CFM | $\rho_0(x_0)\rho_1(x_1)$ | $tx_1 + (1-t)x_0$ | $\sigma$ | VF: (10) |
| Schrödinger Bridge CFM [20] | $\rho_0(x_0)\rho_1(x_1)$ | $tx_1 + (1-t)x_0$ | $\sigma\sqrt{t(1-t)}$ | Can be obtained by SDE using VF: (49), S:(50) |

**Complexity** We assume that the main running time of the algorithm is spent on training the model, especially if it is quite complex. Thus, the running time of one training step depends crucially on the number $n \cdot m$ of samples $\{x\}$ and it is approximately the same for both algorithms: the addition of points $\overline{x}_1$ entails only an additional calculation using formula (16), which can be done quickly and, moreover, can be simple parallelized.

## 2.4 Irreducible dispersion of gradient for CFM optimization

Ensuring the stability of optimization is vital. Let $\Delta\theta$ be changes in parameters, obtained by SGD with step size $\gamma/2$ applied to the functional from Eq. (13):

$$\Delta v(x^{j,i}, t^j) = -\gamma \cdot \left(v(x^{j,i}, t^j) - v^d(x^{j,i}, t^j)\right). \tag{17}$$

For simplification, we consider a function, $v_\theta(x, t)$, capable of perfectly fitting the CFM problem and providing an optimal solution for any point $x$ and time $t$. For a linear conditional flow at a specific point $x^{j,i} \sim \rho_{x_1^i}(\cdot, t^j)$ at time $t^j \sim U(0,1)$, the update $\Delta v(x^{j,i}, t^j)$ can be represented as follows:

$$\Delta v(x^{j,i}, t^j) = \gamma \left(x_1^i - \hat{x}_0^i - v(x^{j,i}, t^j)\right), \tag{18}$$

where $\hat{x}_0^i = \frac{x^{j,i} - t^j x_1^i}{1 - t^j}$. We define the dispersion $\mathbb{D}_{x,x_1} f(x, x_1)$ for $x \sim \rho_{x_1}(\cdot, t)$ and $x_1 \sim \rho_1$ as:

$$\mathbb{D}_{x,x_1} f(x, x_1) = \mathbb{E}_{x,x_1} f^2(x, x_1) - (\mathbb{E}_{x,x_1} f(x, x_1))^2. \tag{19}$$

**Proposition 2.5.** *At the time $t = 0$, the dispersion of update in the form* (18) *have the following element-wise lower bound:*

$$\mathbb{D}_{x^{j,i}, x_1^i} \Delta v(x^{j,i}, 0) = \gamma^2 \mathbb{D}_{x_1^i} x_1^i + \gamma^2 \mathbb{D}_{x^{j,i}, x_1^i}(x^{j,i} + v(x^{j,i}, 0)) \geq \gamma^2 \mathbb{D}_{x_1^i} x_1^i.$$

*Equality is reached when the model $v(x^{j,i}, 0)$ has exact values equal to* (12).

Given that the dispersion cannot be reduced with an increase in batch size, the only available option is to decrease the step size of the optimization method, *i.e.*, reduce the learning rate slowing down the convergence. The situation is much better for the proposed loss in (14). We can express the update $\Delta v(x^{j,i}, t^j)$ in the case of ExFM objective as:

$$\Delta v(x^j, t^j) = \gamma^2 \left(\sum_{k=1}^{N} x_1^k \tilde{\rho}\left(x^{j,i} | x_1^k, t^j\right) - x^{j,i} - v(x^{j,i}, t^j)\right), \tag{20}$$

where $x^{j,i} \sim \rho_{x_1^i}(\cdot, t^j)$, $x_1^k \sim \rho_1$ and $\tilde{\rho}\left(x^{j,i} | x_1^k, t^j\right) = \rho_0\left(\frac{x^{j,i} - t^j x_1^k}{1 - t^j}\right) / \sum_{k=1}^{N} \rho_0\left(\frac{x^{j,i} - t^j x_1^k}{1 - t^j}\right)$. Similar to the derivations in the previous part, we can found simplified form for the dispersion of update at $t = 0$.

**Proposition 2.6.** *At the time $t = 0$, the dispersion of update from* (20) *have the following element-wise lower bound:*

$$\mathbb{D}_{x^{j,i}, x_1^k} \Delta v(x^{j,i}, 0) = \frac{\gamma^2}{N} \mathbb{D}_{x_1^k} x_1^k + \gamma^2 \mathbb{D}_{x^{j,i}, x_1^k}(x^{j,i} + v(x^{j,i}, 0)) \geq \frac{\gamma^2}{N} \mathbb{D}_{x_1^k} x_1^k.$$

*Equality is reached when the model $v(x^{j,i}, 0)$ has exact values equal to* (12).

In comparison to CFM, the dispersion of the update is $N$ times smaller than the dispersion of the target distribution and could be controlled without impeding convergence by adjusting the number of samples $N$. In Figure 1(b), we visually compare the dispersions of CFM and ExFM. The illustration aligns a standard normal distribution $\mathcal{N}(0, I)$ with a shifted and scaled variant $\mathcal{N}(\mu, I\sigma^2)$. ExFM yields lower dispersion throughout the range $t \in [0, 1]$. Detailed analytical calculations of the optimal velocity $v(x, t)$ and dispersion are provided in the Appendix G.

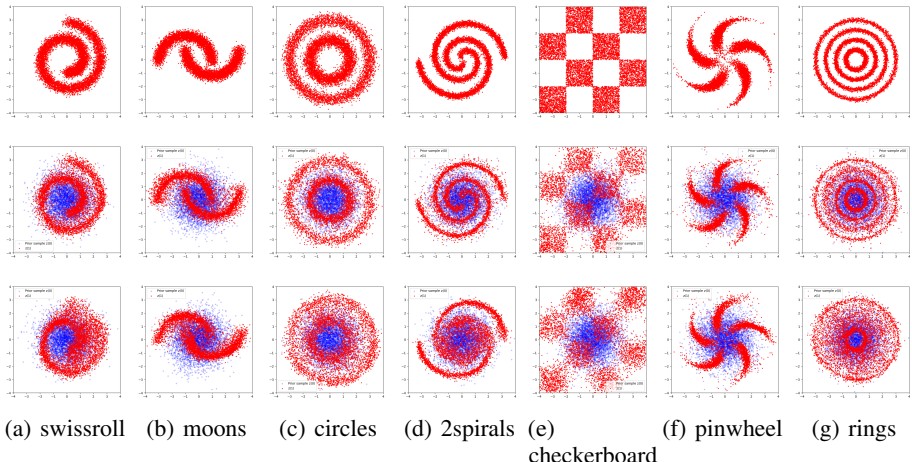

| (a) swissroll | (b) moons | (c) circles | (d) 2spirals | (e) checkerboard | (f) pinwheel | (g) rings |

Figure 3: Visual comparison of methods on toy 2D data. First row are original samples, second row sampled by ExFM, third row sampled by CFM.

Table 2: ExFM and CFM metrics comparison table on toy 2D data.

| DATA | MSE TRAINING LOSS | | ENERGY DISTANCE | |
|---|---|---|---|---|
| | ExFM | CFM | ExFM | CFM |
| SWISSROLL | 1.13E-02 | 2.12E+00 | **2.58e-03** | 1.07E-02 |
| MOONS | 9.96E-03 | 2.01E+00 | **2.74e-03** | 1.41E-02 |
| 8GAUSSIANS | 2.40E-02 | 2.77E+00 | **4.90e-03** | 2.45E-02 |
| CIRCLES | 9.28E-03 | 2.79E+00 | **6.69e-04** | 1.32E-02 |
| 2SPIRALS | 8.92E-03 | 2.34E+00 | **1.27e-03** | 8.35E-03 |
| CHECKERBOARD | 1.04E-02 | 3.12E+00 | **1.01e-02** | 1.63E-02 |
| PINWHEEL | 4.53E-03 | 2.12E+00 | **1.01e-03** | 9.22E-03 |
| RINGS | 8.60E-03 | 1.93E+00 | **3.55e-04** | 2.37E-03 |

Table 3: NLL comparison for ExFM, CFM and OT-CFM methods over 10 000 learning steps, mean and std taken from 10 sampling iterations.

| DATA | ExFM | CFM | OT-CFM |
|---|---|---|---|
| POWER | **-8.51e-02 $\pm$ 4.85e-02** | 1.64E-01 $\pm$ 4.18E-02 | 5.22E-02 $\pm$ 3.92E-02 |
| GAS | **-5.53e+00 $\pm$ 3.66e-02** | -5.00E+00 $\pm$ 2.56E-02 | -5.48E+00 $\pm$ 2.90E-02 |
| HEPMASS | 2.16E+01 $\pm$ 6.31E-02 | **2.21e+01 $\pm$ 6.13e-02** | 2.16E+01 $\pm$ 4.32E-02 |
| BSDS300 | -1.29E+02 $\pm$ 8.40E-01 | -1.29E+02 $\pm$ 8.97E-01 | **-1.32e+02 $\pm$ 6.39e-01** |
| MINIBOONE | **1.34e+01 $\pm$ 1.95e-04** | 1.42E+01 $\pm$ 1.29E-04 | 1.43E+01 $\pm$ 9.22E-05 |

## 3 Numerical Experiments

**Toy 2D data** We conducted unconditional density estimation among eight distributions. Additional details of the experiments see in the Appendix H. We commence the exposition of our findings by showcasing a series of classical 2-dimensional examples, as depicted in Fig. 3 and Table 2. Our observations indicate that ExFM adeptly handles complex distribution shapes is particularly noteworthy, especially considering its ability to do so within a small number of epochs. Additionally, the visual comparison underscores the evident superiority of ExFM over the CFM approach.

**Tabular data**   We conducted unconditional density estimation on five tabular datasets, namely `power`, `gas`, `hepmass`, `minibone`, and `BSDS300`. Additional details of the experiments see in the Appendix H. The empirical findings obtained from the numerical experiments from Table 3 indicate a statistically significant improvement in the performance of our proposed method. Notably, ExFM demonstrates a notable acceleration in convergence rate.

**High-dimensional data and additional experiments**   We conducted experiments on high-dimensional data, among them experiments on CIFAR10 and MNIST dataset. FID results on CIFAR10 shows slightly better score among sampled images.

Additional details of the experiments and sampled images see in the Appendix H.

**Stochastic ExFM (ExFM-S) on toy 2D data**   We evaluated the performance of the stochastic version of ExFM (ExFM-S) with use of expressions given in Sec. E.3.2 on four standard toy datasets. The primary experimental setup follows that used in [19]. Additional details on the hyperparameters used are available in Appendix H. Based on the findings presented in Table 4, we determine that ExFM-S surpasses I-CFM on all four datasets in terms of generative performance ($\mathcal{W}_2$) and also outperforms in terms of OT optimality (NPE) on two of them, exhibiting similar results on the remaining datasets. It also demonstrates performance similar to OT-CFM. While ExFM-S is not as robust as the basic ExFM, it enables the matching of one dataset to another (moons $\to$ 8gaussians) as it does not necessitate the presence of an explicit formula for $\rho_0$. Among other things, this experiment demonstrates the feasibility of our methods when both distributions $\rho_0$ and $\rho_1$ are unknown.

Table 4: ExFM-S evaluation on four toy datasets ($\mu \pm \sigma$ over three seeds). For comparison we take I-CFM, OT-CFM, and ExFM (no values for moons $\to$ 8gaussians due to the absence of explicit formula for $\rho_0$). Performance in generative modeling ($\mathcal{W}_2$) and dynamic OT optimality (NPE) is assessed. The best result for each metric is highlighted in bold. Instances where we outperform CFM are underscored.

| Metric → | $\mathcal{W}_2 \downarrow$ | | | | NPE $\downarrow$ | | | |
|---|---|---|---|---|---|---|---|---|
| Algorithm ↓ Dataset → | $\mathcal{N} \to$ moons | $\mathcal{N} \to$ 8gaussians | moons $\to$ 8gaussians | $\mathcal{N} \to$ 2spirals | $\mathcal{N} \to$ moons | $\mathcal{N} \to$ 8gaussians | moons $\to$ 8gaussians | $\mathcal{N} \to$ 2spirals |
| I-CFM | $0.522 \pm 0.015$ | $0.647 \pm 0.078$ | $0.966 \pm 0.21$ | $1.662 \pm 0.067$ | $0.328 \pm 0.051$ | $0.209 \pm 0.009$ | $0.945 \pm 0.025$ | $0.098 \pm 0.04$ |
| OT-CFM | $0.427 \pm 0.038$ | $0.528 \pm 0.053$ | $\mathbf{0.569 \pm 0.018}$ | $1.322 \pm 0.052$ | $\mathbf{0.065 \pm 0.068}$ | $\mathbf{0.031 \pm 0.018}$ | $\mathbf{0.074 \pm 0.026}$ | $\mathbf{0.031 \pm 0.02}$ |
| ExFM | $\mathbf{0.318 \pm 0.010}$ | $\mathbf{0.445 \pm 0.075}$ | – | $\mathbf{1.276 \pm 0.043}$ | $0.382 \pm 0.050$ | $0.213 \pm 0.023$ | – | $\underline{0.069 \pm 0.064}$ |
| ExFM-S | $0.486 \pm 0.09$ | $0.570 \pm 0.053$ | $0.728 \pm 0.063$ | $1.361 \pm 0.181$ | $0.35 \pm 0.143$ | $\underline{0.166 \pm 0.039}$ | $0.946 \pm 0.059$ | $\underline{0.083 \pm 0.059}$ |

# 4   Conclusions

The presented method introduces a new loss function in tracrable form (in terms of integrals) that improves upon the existing Conditional Flow Matching approach. New loss as a function of the model parameters, reaches zero at its minimum. Thanks to this, we can: a) write an explicit expression for the vector field on which the loss minimum is achieved; b) get a smaller variance when training on the discrete version of the loss, therefore, we can learn the model faster and more accurately.

Numerical experiments conducted on toy 2D data show reliable outcomes under uniform conditions and parameters. Comparison of the absolute values of loss for the proposed method and for CFM for the same distributions show that the absolute values of loss for these models differ strikingly, by a factor of $10^2$–$10^3$. Experiments on high-dimensional datasets also confirm the theoretical deductions about the variance reduction of our method. However, we emphasize that we do not expect to use the proposed method in its pure form. On the contrary, we expect that the theoretical implications of our formulas will contribute to the construction of better learning or inference algorithms in conjunction with other heuristics or methods.

Algebraic analysis of variance for some cases (in particular, for the case $t = 0$ or for the case of two Gaussians as initial and final distributions) show an improvement in variance when using the new loss. However, it is rather difficult to analyze in the general case, for all times $t$ and general distributions $\rho_0$ and $\rho_1$.

Having the expression for the vector field and score in the form of integrals, we can explicitly write out their expressions for some simple cases; in the case of Gaussian distributions we can also write out the exact solution for the trajectories. Thus, our approach allows one to advance the theoretical study of FM-based and Diffusion Model-based frameworks.

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

 # A  Proof of the theorems

 ## A.1  Proof of the Theorem 2.1

 *Proof.* We need to proof, that $\frac{\mathrm{d}L_{\text{CFM}}(\theta)}{\mathrm{d}\theta} = \frac{\mathrm{d}L_{\text{ExFM}}(\theta)}{\mathrm{d}\theta}$.

 To establish the equivalence of $L_{\text{CFM}}$ and $L_{\text{ExFM}}$ up to a constant term, we begin by expressing $L_{\text{CFM}}$
 in the format specified by equation (6):

$$L_{\text{CFM}} = \mathbb{E}_{t,x_1,x\sim\rho_m(\cdot,t)}\|v_\theta(x,\,t) - w(t,x_1,x)\|^2 \times \rho_c(x|x_1,t)/\rho_1(x_1).$$

 Utilizing the bilinearity of the 2-norm, we can rewrite $L_{\text{CFM}}$ as:

$$L_{\text{CFM}} = \mathbb{E}_{t,x_1,x\sim\rho_m(\cdot,t)}\frac{\|v_\theta(x,\,t)\|^2\rho_c(x|x_1,t)}{\rho_1(x_1)} -$$
$$2\mathbb{E}_{t,x_1,x\sim\rho_m(\cdot,t)}\frac{v_\theta(x,\,t)^T \cdot w(t,x_1,x)\rho_c(x|x_1,t)}{\rho_1(x_1)} + C. \quad (21)$$

 Here, $T$ denotes transposed vector, dot denotes scalar product, $C$ represents a constant independent
 of $\theta$.

 Noting that $\mathbb{E}_{x_1}\rho_c(x|x_1,t)/\rho_1(x_1) = 1$:

$$\mathbb{E}_{x_1}\frac{\rho_c(x|x_1,t)}{\rho_1(x_1)} = \int \frac{\rho_{x_1}(x,t)\rho_1(x_1)\,\mathrm{d}x_1}{\int \rho_{x_1}(x,t)\rho_1(x_1)\,\mathrm{d}x_1} = 1,$$

 we can simplify the first term in the expansion (21):

$$\mathbb{E}_{t,x_1,x\sim\rho_m(\cdot,t)}\frac{\|v_\theta(x,\,t)\|^2\rho_c(x|x_1,t)}{\rho_1(x_1)} =$$
$$E_{t,x\sim\rho_m(\cdot,t)}\|v_\theta(x,\,t)\|^2 \, \mathbb{E}_{x_1}\frac{\rho_c(x|x_1,t)}{\rho_1(x_1)} = E_{t,x\sim\rho_m(\cdot,t)}\|v_\theta(x,\,t)\|^2. \quad (22)$$

 For our loss $L_{\text{ExFM}}$ in the form (8) we also use the bilinearity of the norm:

$$L_{\text{ExFM}} = \mathbb{E}_{t,x\sim\rho_m(\cdot,t)}\|v_\theta(x,\,t)\|^2 - 2\mathbb{E}_{t,x\sim\rho_m(\cdot,t)}\mathbb{E}_{x_1}\frac{v_\theta(x,\,t)^T \cdot w(t,x_1,x)\rho_c(x|x_1,t)}{\rho_1(x_1)} + C. \quad (23)$$

 Comparing the last expression and the Eq. (21) with the modification (22) and also taking into account
 the independence of random variables $x$ and $x_1$, we come to the conclusion that $L_{\text{ExFM}}$ is equal to
 $L_{\text{CFM}}$ up to some constant independent of the model parameters.

 $\square$

 ## A.2  Sketch of the proof of the Theorem 2.4

 *Proof.* We need to prove that $\mathbb{D}\frac{\mathrm{d}L_{\text{ExFM}}^d(\theta)}{\mathrm{d}\theta} \leq \mathbb{D}\frac{\mathrm{d}L_{\text{CFM}}^d(\theta)}{\mathrm{d}\theta}$, where $L_{\text{ExFM}}^d(\theta)$ and $L_{\text{CFM}}^d(\theta)$ discrete loss
 functions presented in (14) and (13). Firstly, let us rewrite the derivative of loss functions using the
 bilinearity:

$$\frac{\mathrm{d}L_{\text{ExFM}}^d(\theta)}{\mathrm{d}\theta} = 2\sum_{i,j}\left(\frac{\mathrm{d}v_\theta(x^{j,i},\,t^j)}{\mathrm{d}\theta}\right)^T \cdot \left(v_\theta(x^{j,i},\,t^j) - v^d(x^{j,i},\,t^j)\right).$$

 Note that in this expression, values $x^{j,i}$ as well as $t^j$, which are included in the argument of the
 function $v$, are fixed (our goal to calculate the variance with fixed model arguments). Thus, we need
 to consider the variance of the remaining expression arising from the randomness of $\overline{x}_1^k$.

Recall (below we will omit the indices at variables $x$ and $t$),

$$v^d(x,\,t) = \frac{\sum_{k=1}^N w(t,\overline{x}_1^k,x) \cdot \rho_0\big(\phi_{t,\overline{x}_1^k}^{-1}(x)\big)}{\sum_{k=1}^N \rho_0\big(\phi_{t,\overline{x}_1^k}^{-1}(x)\big)}.$$

Note, that if $N = 1$, (*i. e.* we do not sample any additional points other than the ones we have already sampled) this expression is exactly the same as the derivative of the common discretized CFM loss $\frac{\mathrm{d}L_{\mathrm{CFM}}^d(\theta)}{\mathrm{d}\theta}$.

Moreover, recall that one of the points (without loss of generality, we can assume that its index is 1) $\overline{x}_1^1$ is added from the set from which point $x$ was derived: $x = \phi_{t,\overline{x}_1^1}(x_0)$. (Here $x_0$ is the paired point to $\overline{x}_1^1$)

Thus, we can rewrite expression for $v^d$:

$$v^d(x,\,t) = \frac{w(t, \overline{x}_1^1, x)\rho_0(x_0) + \sum_{k=2}^N w(t, \overline{x}_1^k, x) \cdot \rho_0\left(\phi_{t,\overline{x}_1^k}^{-1}(x)\right)}{\rho_0(x_0) + \sum_{k=2}^N \rho_0\left(\phi_{t,\overline{x}_1^k}^{-1}(x)\right)}. \tag{24}$$

Thus, our task was reduced to evaluating how well the additional terms (for $k$ starting from 2) improve approximate of the original integrals that are in loss (8).

So, we need to estimate the following dispersion ratio, where in the numerator is the variance of discrete loss CFM, and in the denominator — the variance of loss ExFM:

$$k_D = \frac{\mathbb{D}\left(v_\theta(x,t) - w(t, \overline{x}_1^1, x)\right)}{\mathbb{D}\left(v_\theta(x,t) - \dfrac{\sum_{k=1}^N w(t,\overline{x}_1^k,x)\cdot\rho_0\left(\phi_{t,\overline{x}_1^k}^{-1}(x)\right)}{\sum_{k=1}^N \rho_0\left(\phi_{t,\overline{x}_1^k}^{-1}(x)\right)}\right)}$$

The smaller coefficient $k_D$ is, the better the proposed loss ExFM works.

Formally, we can write our problem as an importance sampling problem for the following integral:

$$I = \int f(x)p(x)\,\mathrm{d}x\,.$$

This integral we estimate by sample mean of the following expectation over some random variable with density function $q(x)$:

$$I = \mathbb{E}_{x \sim q}\left(w(x)f(x)\right)$$

with

$$w(x) = \frac{p(x)}{q(x)}.$$

We replace the exact value of $I$ with the value

$$\overline{I} = \frac{\sum_{k=1}^N w(\overline{x}_1^i)f(\overline{x}_1^k)}{\sum_{i=k}^N w(\overline{x}_1^k)}.$$

It follows from the strong law of large numbers that in the limit $N \to \infty$, $I \to \overline{I}$ almost surely. From the central limit theorem we can find the asymptotic variance:

$$\mathbb{D}\overline{I} = \frac{1}{N}\mathbb{E}_{x \sim q}\left(w^2(x)(f(x) - I)^2\right). \tag{25}$$

In our case (loss $L_{\mathrm{ExFM}}$), we have $q(x_1) = \rho_1(x_1)$, $f(x_1) = w(t, x_1, x)$ and $w(x_1) = \rho_0\left(\phi_{t,x_1}^{-1}(x)\right)$.

Despite the fact that the equation (25) for the variance contains $N$ in the denominator, it is rather difficult to give an estimate of its behavior in general. The point is that this formula is well suited for the case when $w$ in it is of approximately the same order. In the considered case, this is achieved at times $t$ noticeably less than 1.

But in the case, when $t$ is closed to 1 we have, for example, for the linear map, that

$$w(x_1) = \rho_0\left(\phi_{t,x_1}^{-1}(x)\right) = \rho_0\left(\frac{x - x_1 t}{1 - t}\right)$$

and this function has a sharp peak near the point $x/t$ if it is considered as a function of $x_1$. Thus, at such values of $t$, only a small number of summands will give a sufficient contribution to the sum compared to the first term.

Finally, inequality $k_D < 1$ is formally fulfilled, but how much $k_D$ is less than one depends on many factors.

$\square$

## A.3 Expressions for the regularized map

To justify the expression (11), we use a invertable transformation and then strictly take the limit $\sigma_s \to 0$.

Expression Eq. (11), (16) are obtained for the simple map $\phi_{t,x_1}(x_0) = (1-t)x_0 + tx_1$ which is not invertable at $t = 1$. For the map with small regaluraziting parameter $\sigma_s > 0$ $\phi_{t,x_1}(x_0) = (1-t)x_0 + tx_1 + \sigma_s x_0$, which is invertable at all time values $0 \le t \le 1$, Eq. (11), (16) needs modifications. Namely, for this map the following exact formulas holds true

$$v(x,t) = \int w(t,x_1,x)\rho_c(x|x_1,t)\rho_1(x_1)\,\mathrm{d}x_1 = \frac{\int \left(x_1 - x(1-\sigma_s)\right)\rho_0\left(\frac{x-x_1 t}{1+\sigma_s t - t}\right)\rho_1(x_1)\,\mathrm{d}x_1}{(1+\sigma_s t - t)\int \rho_0\left(\frac{x-x_1 t}{1+\sigma_s t - t}\right)\rho_1(x_1)\,\mathrm{d}x_1}.$$

(26)

By direct substitution we make sure that for this vector field

$$v(x,\,0) = \int x_1 \rho_1(x_1)\,\mathrm{d}x_1 - x(1-\sigma_s)$$

(27)

and

$$v(x,\,1) = \frac{\int (x-y)\rho_0(y)\rho_1(x-y\sigma_s)\,\mathrm{d}y}{\int \rho_0(y)\rho_1(x-y\sigma_s)\,\mathrm{d}y},$$

(28)

where we perform change of the variables $y \leftarrow \frac{x_1-x}{\sigma_s t}$.

### A.3.1 Prof of the explicit formula (11) for the vector field

**Assumption A.1.** Density $\rho_1$ is continuous at any point $x \in (-\infty,\,\infty)$.

**Theorem A.2.** *In equations* (26), (27) *and* (28) *we can take the limit* $\sigma_s \to 0$ *under integrals to get Eq.* (11) *and* (12).

*Proof.* Assuming that the distribution $\rho_1$ has a finite first moment: $|\int \xi \rho_1(\xi)\,\mathrm{d}\xi| < C_1$ and that the density of $\rho_0$ is bounded: $\rho_0(x) < C_2$, $\forall x \in (-\infty,\,\infty)$, we obtain that the integrand functions in the numerator and denominator in the Eq. (26) can be bounded by the following integrable functions independent of $\sigma_s$ and $t$:

$$\rho_0\left(\frac{x-x_1 t}{1+\sigma_s - t}\right)\rho_1(x_1) < C_1 \rho_1(x_1)$$

and

$$0 \le x_1 \rho_0\left(\frac{x-x_1 t}{1+\sigma_s t - t}\right)\rho_1(x_1) < x_1 C_1 \rho_1(x_1), \quad x \ge 0,$$

$$0 > x_1 \rho_0\left(\frac{x-x_1 t}{1+\sigma_s t - t}\right)\rho_1(x_1) > x_1 C_1 \rho_1(x_1), \quad x < 0.$$

It follows that both integrals in expression (26) converge absolutely and uniformly. So, we can swap the operations of taking the limit and integration, and we can take the limit $\sigma_s \to 0$ in the integrand for any time $t \in [0,\,t_0]$ for arbitrary $t_0 < 1$.

Now, let us consider the case $t = 1$. From Assumption A.1 the boundedness of the density $\rho_1$ follows: $\rho_1(x) < C_2$, $\forall x \in (-\infty,\,\infty)$. Thus, integrand functions in the numerator and denominator in the Eq. (28) can be bounded by the following integrable functions independent of $\sigma_s$:

$$\rho_0(y)\rho_1(x-y\sigma_s) < \rho_0(y)C_2$$

and

$$0 \le y\rho_0(y)\rho_1(x-y\sigma_s) < yC_2\rho_0(y), \quad y \ge 0,$$
$$0 > y\rho_0(y)\rho_1(x-y\sigma_s) > yC_2\rho_0(y), \quad y < 0.$$

The existence of the limit

$$\lim_{\sigma_s \to 0} \rho_1(x-y\sigma_s) = \rho_1(x),$$

follows from Assumption A.1.

Finally, we conclude that formula (11), regarded as the limit $\sigma_s \to 0$ of the (26) at any $t \in [0,\,1]$, is true. $\qquad\square$

**Theorem A.3.** *The vector field in Eq. (11) delivers minimum to the Flow Matching objective (see the work [9]),*

$$\mathbb{E}_t \mathbb{E}_{x \sim \rho(x,t)} \| \overline{v}(x,t) - v(x,t) \|,$$

*where $\rho(x,t)$ and $\overline{v}(x,t)$ satisfy the equation* (1) *with the given densities $\rho_0$ and $\rho_1$.*

*Proof.* The proof is based on the previous statements and on a Theorem 1 from [9] (that the marginal vector field based on conditional vector fields generates the marginal probability path based on conditional probability paths.

To complete the proof, we must justify that, with $\sigma_s$ tending to zero, the marginal path at $t = 1$ coincides with a given probability $\rho_1$.

Consider the marginal probability path $p_t(x,t)$

$$p_t(x,t) = \int p_t(x|x_1, \sigma_s)\rho_1(x_1)\mathrm{d}x_1 \tag{29}$$

where $p_t(x|x_1, \sigma_s)$ is conditional probability paths obtained by regularized linear conditional map. Distribution $p_t$ in the time $t = 0$ is equal to standard normal distribution $p_0(x|x_1, \sigma_s) = \mathcal{N}(x \mid 0, 1)$ and at the time $t = 1$ it is a stretched Gaussian centered at $x_1$: $p_1(x|x_1, \sigma_s) = \mathcal{N}(x \mid x_1, \sigma_s I)$.

Substituting $p_1$ into the Eq. (29) and considering that there exists a limit $\sigma_s \to 0$ due to Assumption A.1, we obtain

$$p_1(x) = \lim_{\sigma_s \to 0} \int p_t(x|x_1, \sigma_s)\rho_1(x_1)\mathrm{d}x_1 = \rho_1(x_1).$$

This finish the proof. $\qquad\square$

### A.3.2 Learning procedure for $\sigma_s > 0$

Using standard normal distribution as initial density $\rho_0$, and the regularized map $\phi_{t,x_1}(x_0) = (1-t)x_0 + tx_1 + \sigma_s t x_0$ we obtain the following approximation formula

$$v^d(x,\,t) = \frac{\sum_{k=1}^{N} \frac{\overline{x}_1^k - x(1-\sigma_s)}{1 - t(1-\sigma_s)} \exp\left(Y^k\right)}{\sum_{k=1}^{N} \exp(Y^k)}, \quad \text{where} \quad Y^k = -\frac{1}{2}\frac{\left\| x - t \cdot \overline{x}_1^k \right\|_{\mathbb{R}^d}^2}{1 - t(1-\sigma_s)}.$$

In practical applications, the exponent calculation is replaced by the SoftMax function calculation, which is more stable.

## B  Estimation of integrals

In general, we need to estimate the following expression

$$I(\eta) = \frac{\int w(x_1,\, \eta)f(x_1,\, \eta)\rho_1(x_1)\,\mathrm{d}x_1}{\int f(x_1,\, \eta)\rho_1(x_1)\,\mathrm{d}x_1}.$$

In particular, substituting $\eta \to \{x,t\}$, $w(x,\eta) \to (x_1 - x)/(1-t)$ we obtain formula (11) and similar ones with similar substitutions.

If we can sample from the $\rho_1$ distribution, we can estimate this integral in two ways: *self-normalized importance sampling* and *rejection sampling*.

Let $\mathcal{X} = \{x_1^k\}_{k=1}^{N}$ be $N$ samples from the distribution $\rho_1$.

**Self-normalized Importance Sampling**  In this case

$$I(\eta) \approx \frac{\sum\limits_{k=1}^{N} w(x_1^k, \eta)f(x_1^k, \eta)\rho_1(x_1^k)\,\mathrm{d}x_1}{\sum\limits_{k=1}^{N} f(x_1^k, \eta)\rho_1(x_1^k)\,\mathrm{d}x_1}. \tag{30}$$

This estimate is biased in theory, but there several methods to reduce this bias and improve this estimate, see, for example, [3]. Our numerical experiments generally show that the estimation (30) in the form is already sufficient for stable results; we don not observe any bias.

**Rejection sampling**  Let $\mathcal{Y} = \{y^k\}_{k=1}^{M} \subset \mathcal{X}$ be a subset of the the initially given set of samples, which is formed according to the following rule. Let $C = \sup_x \rho_1(x)$. For a given sample $x_1^j$ we generate a random uniformly distributed variable $\xi_j \sim \mathcal{U}(0,1)$ and if

$$f(x_1^j) \geq C\xi_j,$$

then we put the point $x_k^j$ to the set $\mathcal{Y}$; otherwise we reject it.

Having formed the set $\mathcal{Y}$, we evaluate the integral as

$$I(\eta) \approx \frac{1}{M} \sum_{k=1}^{M} w(y^k, \eta).$$

To justify the last estimation, we note, that the points from the set $\mathcal{Y}$ are distributed according to (non-normalized) density $\rho(x)f(x,\eta)\rho_1(x)$. One can show it using the proof of the rejection sampling method. This is the same density as in Eq. (7) and thus we estimate the expression (10) using Important Sampling without any additional denominator.

**Comparison**  When we apply these techniques to evaluating the expression for the vector field, we know that when the time parameter $t$ is close to 1, the function $f(x_1, \eta)$ (which is a scaled $\rho_0$) has a peak at the point $x = x_1$. This means that only a small number of points from the original set will end up in the set $\mathcal{Y}$. Moreover, in the case when the time $t$ is very close to one and the data are well separated, only one point $x_1$ will end up in $\mathcal{Y}$. This explains why we initially put this point in the set $\mathcal{X}$, because otherwise it would be possible that the set $\mathcal{Y}$ is empty and $M = 0$.

As a future work, we indicate a theoretical finding of the probability of hitting a particular point $x_1$ in the set $\mathcal{Y}$ and, thus, a modification of our algorithm, when the sample $x_1$ will not always go to the set $\mathcal{X}$, but with some probability — the greater the $t$ the closer this probability to 1.

## C  The main Algorithm and extensions and generalization of the exact expression

---
**Algorithm 1** Vector field model training algorithm

---
**Require:** Sampler from distribution $\rho_1$ (or a set of samples); parameters $n$ and $m$ (number of spatial and time points, correspondingly); parameter $N$ (number of averaging point); model $v_\theta(x,t)$; algorithm with parameters for SGD
**Ensure:** quasi-optimal parameters $\theta$ for the trained model
  1: Initialize $\theta$ (maybe random)
  2: **while** exit condition is not met **do**
  3:    Sample $m$ points $\{t^j\}$ from $\mathcal{U}[0,1]$
  4:    Sample $n$ points pairs $\{x_0^i, x_1^i\}_{i=1}^{n}$ from joint distribution $\pi$ ($\pi(x_0, x_1) = \rho_0(x_0)\rho_1(x_1)$ if variables are independent)
  5:    Sample $N-n$ points $\{\hat{x}_1^l\}$ from $\rho_1$ and form $\{\overline{x}_1^k\} = \{x_1^i\} \cup \{\hat{x}_1^l\}$ // *We can take all available samples as $\{\overline{x}_1^k\}$ if we don't have access to a sampler, but only ready-made samples.*
  6:    For all $i$ and $j$ calculate the sum at the right side of (14) (using (16) if $\rho_0$ is standard Gaussian or (24) in general)
  7:    Calculate the sum on $i$ and $j$ in discrete loss (14), and take backward derivative, obtaining approximate grad $G \approx \nabla_\theta L_{\text{ExFM}}$ of loss $L_{\text{ExFM}}$ on model parameters $\theta$.
  8:    Update model parameters $\theta \leftarrow SGD(\theta, G)$
  9: **end while**

---

General form of the proposed Algorithm is given in Alg 1.

When using other maps, formula (11) is modified accordingly. For example, if we use the regularized map $\phi_{t,x_1}(x_0) = (1-t)x_0 + tx_1 + \sigma_s t x_0$, we get the formula (26).Note, that in this case the final density $\rho(x,1)$, obtained from the continuity equation is not equal to $\rho_1$, but is its smoothed modification.

When using a different initial density $\rho_0$ (not the normal distribution), an obvious modification will be made to formula (16).

**Diffusion-like models** We can treat so-called Variance Preserving [6] model as CFM with the map

$$\phi_{t,x_1}(x) = \alpha_{1-t}x + \sqrt{1-\alpha_{1-t}^2}\, x_1.$$

and $\rho_0$ as standard normal distribution: $\rho_0 = \mathcal{N}\left(\cdot\,\middle|\,0, 1^2\right)$ In this case, the common expression (10) for vector filed transforms to

$$v(x,t) = \frac{\int (x\alpha_{1-t} - x_1)\alpha_{1-t}'\,\rho_0\left(\frac{x - x_1\alpha_{1-t}}{\sqrt{1-\alpha_{1-t}^2}}\right)\rho_1(x_1)\,\mathrm{d}x_1}{(1-\alpha_{1-t}^2)\int \rho_0\left(\frac{x - x_1\alpha_{1-t}}{\sqrt{1-\alpha_{1-t}^2}}\right)\rho_1(x_1)\,\mathrm{d}x_1}, \tag{31}$$

where $\alpha_s' = \frac{\mathrm{d}\alpha_s}{\mathrm{d}s}$.

Similarity we can treat so-called Variance Exploding [17] model as CFM with the map

$$\phi_{t,x_1}(x) = \sigma_{1-t}x + x_1.$$

and $\rho_0$ also as standard normal distribution: $\rho_0 = \mathcal{N}\left(\cdot\,\middle|\,0, 1^2\right)$ In this case, the common expression (10) for vector filed transforms to

$$v(x,t) = \frac{\int (x_1 - x)\sigma_{1-t}'\,\rho_0\left(\frac{x - x_1}{\sigma_{1-t}}\right)\rho_1(x_1)\,\mathrm{d}x_1}{\sigma_{1-t}\int \rho_0\left(\frac{x - x_1}{\sigma_{1-t}}\right)\rho_1(x_1)\,\mathrm{d}x_1}, \tag{32}$$

where $\sigma_s' = \frac{\mathrm{d}\sigma_s}{\mathrm{d}s}$.

**Joint Distribution** Moreover, in addition to the independent densities $x_0 \sim \rho_0$ and $x_1 \sim \rho_1$, we can use the joint density $\{x_0, x_1\} \sim \pi(x_0, x_1)$. In the papers [20, 19], optimal transport (OT) and Schrödinger's bridge are taken as $\pi$. In this case the expression for the vector field changes insignificantly: the conditional probability $\rho_c$ from Eq. (7) is subject to change:

$$\rho_c(x|x_1, t) = \frac{\pi\left(\phi_{t,x_1}^{-1}(x), x_1\right)\det\left[\frac{\partial \phi_{t,x_1}^{-1}(x)}{\partial x}\right]}{\int \pi\left(\phi_{t,x_1}^{-1}(x), x_1\right)\det\left[\frac{\partial \phi_{t,x_1}^{-1}(x)}{\partial x}\right]\mathrm{d}x_1}. \tag{33}$$

Then, Eq. (10) remains the same in general case. In the case of linear $\phi$, the extension of Eq. (11) reads

$$v(x,t) = \frac{\int (x_1 - x)\,\pi\left(\phi_{t,x_1}^{-1}(x), x_1\right)\det\left[\frac{\partial \phi_{t,x_1}^{-1}(x)}{\partial x}\right]\mathrm{d}x_1}{(1-t)\int \pi\left(\phi_{t,x_1}^{-1}(x), x_1\right)\det\left[\frac{\partial \phi_{t,x_1}^{-1}(x)}{\partial x}\right]\mathrm{d}x_1}. \tag{34}$$

In all of the above cases, the essence of Algorithm 1 does not change (except that in the case of dependent $x_0$ and $x_1$ we should be able either to calculate the value of $\pi\left(\phi_{t,x_1}^{-1}(x), x_1\right)/\rho_1(x_1)$ or to estimate it).

# D  Several analytical results, following from the explicit formula

In this section, we present several analytical results that directly follow from our exact formulas for the vector field, which, to the best of our knowledge, have not been published before.

## D.1  Exact path from one Gaussian to another Gaussian

Consider the flow from a one-dimensional Gaussian distribution $\rho_0 \sim \mathcal{N}\left(\cdot\,\middle|\,\mu_0, \sigma_0^2\right)$ into another (with other parameters) Gaussian distribution $\rho_1 \sim \mathcal{N}\left(\cdot\,\middle|\,\mu_1, \sigma_1^2\right)$. Note that in this case the generalization to the multivariate case is done directly, so the spatial variables are separated.

From the general formula (11) we have:

$$v(x, t) = \frac{\int (x_1 - x) \mathcal{N}\left(\frac{x - tx_1}{1-t} \middle| \mu_0, \sigma_0^2\right) \mathcal{N}\left(x_1 \middle| \mu_1, \sigma_1^2\right) dx_1}{(1-t) \int \mathcal{N}\left(\frac{x - tx_1}{1-t} \middle| \mu_0, \sigma_0^2\right) \mathcal{N}\left(x_1 \middle| \mu_1, \sigma_1^2\right) dx_1}. =$$

$$= \frac{\int (x_1 - x) \exp\left(-\left(\frac{x - tx_1}{1-t} - \mu_0\right)^2/(2\sigma_0^2) - (x_1 - \mu_1)^2/(2\sigma_1^2)\right) dx_1}{(1-t) \int \exp\left(-\left(\frac{x - tx_1}{1-t} - \mu_0\right)^2/(2\sigma_0^2) - (x_1 - \mu_1)^2/(2\sigma_1^2)\right) dx_1}.$$

Both integrals in the last expression are taken explicitly:

$$\int \mathcal{N}\left(\frac{x - tx_1}{1-t} \middle| \mu_0, \sigma_0^2\right) \mathcal{N}\left(x_1 \middle| \mu_1, \sigma_1^2\right) dx_1 =$$

$$= \frac{\exp\left(-\frac{(x - \mu_0(1-t) - \mu_1 t)^2}{2\left(\sigma_1^2 t^2 + \sigma_0^2(1-t)^2\right)}\right)}{\sqrt{2\pi}\sqrt{\sigma_0^2 + \frac{\sigma_1^2 t^2}{(t-1)^2}}} = \mathcal{N}\left(\frac{x}{1-t} \middle| \frac{\mu_0(1-t) + \mu_1 t}{1-t}, \sigma_0^2 + \frac{\sigma_1^2 t^2}{(t-1)^2}\right).$$

Note that the last relation can be obtained as a distribution of two Gaussian random variables with corresponding parameters.

The second integral:

$$\int \frac{x_1 - x}{1-t} \mathcal{N}\left(\frac{x - tx_1}{1-t} \middle| \mu_0, \sigma_0^2\right) \mathcal{N}\left(x_1 \middle| \mu_1, \sigma_1^2\right) dx_1 =$$

$$= \frac{\exp\left(-\frac{(x - \mu_0(1-t) - \mu_1 t)^2}{2\left(\sigma_1^2 t^2 + \sigma_0^2(1-t)^2\right)}\right)}{\sqrt{2\pi}} \frac{(1-t)\left(\sigma_1^2 t(x - \mu_0) + \sigma_0^2(t-1)(x - \mu_1)\right)}{\left(\sigma_1^2 t^2 + \sigma_0^2(1-t)^2\right)^{3/2}}.$$

Thus, in the considered case we can explicitly write the expression for the vector field $v$:

$$v(x, t) = \frac{\sigma_1^2 t(x - \mu_0) - \sigma_0^2(1-t)(x - \mu_1)}{\sigma_1^2 t^2 + \sigma_0^2(1-t)^2}. \tag{35}$$

For this vector field we can explicitly solve the equation for the path $x(t)$ starting from the arbitrary point $x_0$

$$\begin{cases} \dfrac{\partial x(t)}{\partial t} = v(x(t), t), \\ x(0) = x_0 \end{cases}.$$

The solution is:

$$x(t) = (1-t)\mu_0 + t\mu_1 + (x_0 - \mu_0)\sqrt{(\sigma_1/\sigma_0)^2 t^2 + (1-t)^2}. \tag{36}$$

Note that although this solution does not correspond to the Optimal Transport joint distribution, since the obtained path is not a straight line in general, (*i. e.* we do not have a solution to the Kantorovich's formulation of the OT problem) the endpoint $x(1) = \mu_1 + (x_0 - \mu_0)\dfrac{\sigma_1}{\sigma_0}$ falls exactly in the one that is optimal if we solve the OT problem in the Monge formultation. Thus, the map $x(0) \to x(1)$ is the OT map for the case of 2 Gaussian.

See the Fig. 4 for the examples of the paths for the obtained solution.

## D.2 From one Gaussian to Gaussian Mixture

Let initial distribution be standard Gaussian $\rho_0 = \mathcal{N}\left(\cdot \middle| 0, 1^2\right)$, and the target distribution be Gaussian Mixture (GM) of two symmetric Gaussians: $\rho_1(x) = 1/2(\mathcal{N}\left(x \middle| \mu, \sigma^2\right)) + \mathcal{N}\left(x \middle| -\mu, \sigma^2\right))$, In this

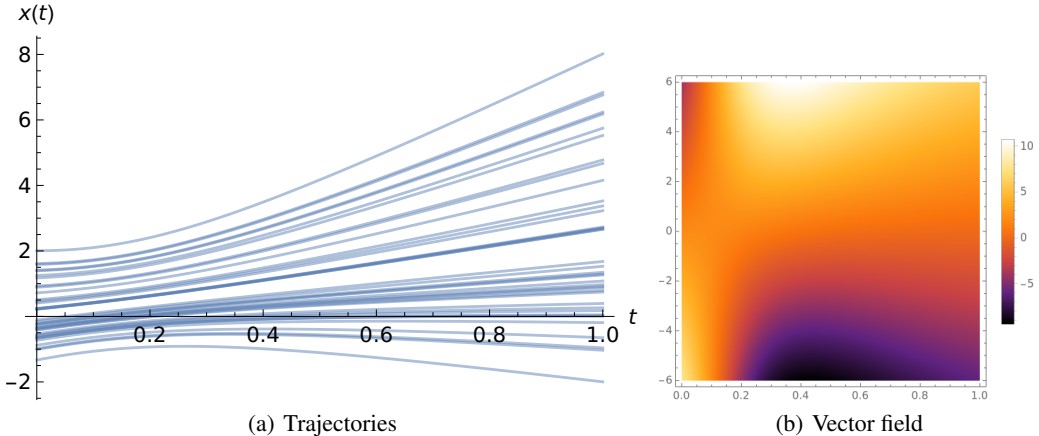

(a) Trajectories
(b) Vector field

Figure 4: a) $N = 40$ random trajectories from from $\mathcal{N}\left(\cdot \,\middle|\, 0, 1^2\right)$ to $\mathcal{N}\left(\cdot \,\middle|\, 2, 3^2\right)$; (b) 2D plot of the vector field in this case

case, we can obtain exact form for $v$

$$
v(x,t) = \frac{\exp\left(-\frac{\mu^2}{2\sigma^2} + \frac{\mu^2 t^2 + x^2}{\sigma^2 t^2 + (t-1)^2} - \frac{x^2}{2(t-1)^2}\right)}{(\sigma^2 t^2 + (t-1)^2)\left(e^{\frac{(x-\mu t)^2}{2(\sigma^2 t^2 + (t-1)^2)}} + e^{\frac{(\mu t + x)^2}{2(\sigma^2 t^2 + (t-1)^2)}}\right)} \times
$$

$$
\left[\mu(t-1)\left(\exp\left(\frac{\left(\mu(t-1)^2 - \sigma^2 tx\right)^2}{2\sigma^2(t-1)^2\left(\sigma^2 t^2 + (t-1)^2\right)}\right) - \exp\left(\frac{\left(\mu(t-1)^2 + \sigma^2 tx\right)^2}{2\sigma^2(t-1)^2\left(\sigma^2 t^2 + (t-1)^2\right)}\right)\right) + \right.
$$

$$
\left. + x\left(\sigma^2 t + t - 1\right)\left(\exp\left(\frac{\left(\mu(t-1)^2 - \sigma^2 tx\right)^2}{2\sigma^2(t-1)^2\left(\sigma^2 t^2 + (t-1)^2\right)}\right) + \exp\left(\frac{\left(\mu(t-1)^2 + \sigma^2 tx\right)^2}{2\sigma^2(t-1)^2\left(\sigma^2 t^2 + (t-1)^2\right)}\right)\right)\right],
\tag{37}
$$

but the expression for the path $x(t)$ is unknown.

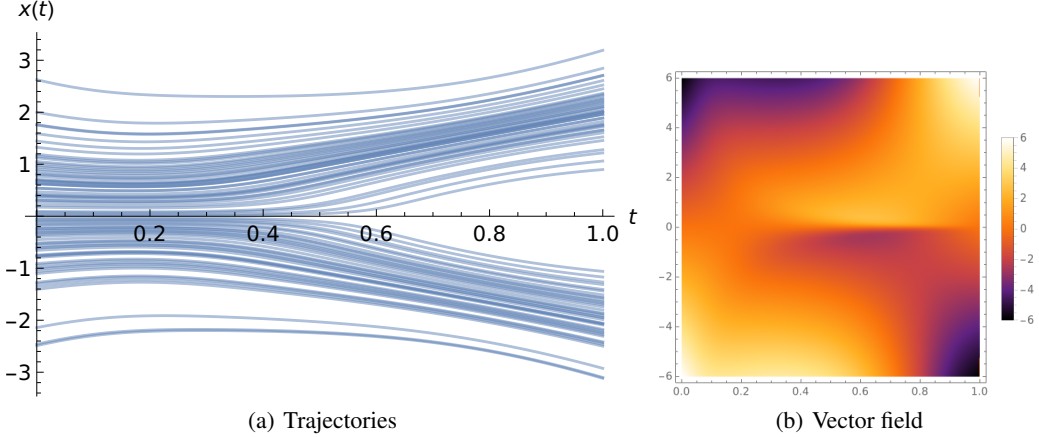

(a) Trajectories
(b) Vector field

Figure 5: a) $N = 80$ random trajectories from $\mathcal{N}\left(\cdot \,\middle|\, 0, 1^2\right)$ to GM of $\mathcal{N}\left(\cdot \,\middle|\, -2, 1/2^2\right)$ and $\mathcal{N}\left(\cdot \,\middle|\, 2, 1/2^2\right)$; (b) 2D plot of the vector field in this case

Numerically solution of the differential equation with the obtained vector field give the trajectories shown in Fig. 5.

### D.3 From Gaussian to Gaussian with stochastic

Using Eq. (44)-(46) we can explicitly calculate vector field $v$ and score $s$ with the setup as in Sec. D.1 but with additional noise, *i. e.* in the stochastic case.

#### D.3.1 Gaussian to Gaussian with noise

Consider like in the Sec. D.1 the flow from a one-dimensional standard Gaussian distribution $\rho_0 \sim \mathcal{N}\left(\cdot \,\middle|\, 0, 0^2\right)$ into another (with other parameters) Gaussian distribution $\rho_1 \sim \mathcal{N}\left(\cdot \,\middle|\, \mu_1, \sigma_1^2\right)$ but with additional noise as described above.

In this case we have for the field.

$$v(x,\,t) = \frac{x\left(t\sigma_1^2 + (1-t)\sigma_e^2/2\right) - (x-\mu_1)\left((1-t) + t\sigma_e^2/2\right)}{t(1-t)\sigma_e^2 + \sigma_1^2 t^2 + (1-t)^2} \tag{38}$$

We can solve ODE with this field and get the expression for the trajectories, starting from the given point $x_0$:

$$x(t) = \mu_1 t + x_0 \sqrt{t(1-t)\sigma_e^2 + \sigma_1^2 t^2 + (1-t)^2}. \tag{39}$$

These trajectories, for different $x_0$ are depicted in Fig. 6.

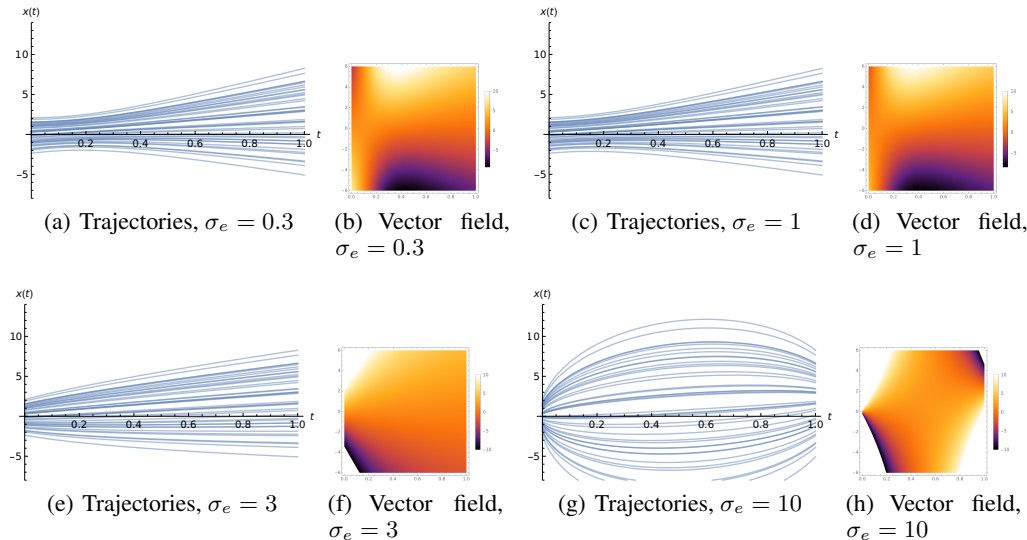

(a) Trajectories, $\sigma_e = 0.3$  (b) Vector field, $\sigma_e = 0.3$  (c) Trajectories, $\sigma_e = 1$  (d) Vector field, $\sigma_e = 1$

(e) Trajectories, $\sigma_e = 3$  (f) Vector field, $\sigma_e = 3$  (g) Trajectories, $\sigma_e = 10$  (h) Vector field, $\sigma_e = 10$

Figure 6: a) $N = 40$ random trajectories from $\mathcal{N}\left(\cdot \,\middle|\, 0, 1^2\right)$ to $\mathcal{N}\left(\cdot \,\middle|\, 2, 3^2\right)$ and 2D plot of the vector field in this case for different $\sigma_e$

At the limit $\sigma_e \to 0$ expressions (38) and (39) turn into expressions (35) and (36) as expected.

For the score $s$ in the considered case we have

$$s(x,\,t) = \frac{t\mu_1 - x}{(1-t)^2 + t(1-t)\sigma_e^2 + t^2\sigma_1^2}$$

Thus, we can explicitly write expressions for the stochastic process for the evolution from the initial distribution $rho_0$ (standard Gaussian) to the final distribution $\rho_1$:

$$\mathrm{d}x(x) = \left[\frac{x\left(t\sigma_1^2 + (1-t)\sigma_e^2/2\right) - (x-\mu_1)\left((1-t) + t\sigma_e^2/2\right)}{t(1-t)\sigma_e^2 + \sigma_1^2 t^2 + (1-t)^2} + \right.$$

$$\left. + \frac{g^2(t)}{2}\frac{t\mu_1 - x}{(1-t)^2 + t(1-t)\sigma_e^2 + t^2\sigma_1^2}\right]\mathrm{d}t + g(t)\,\mathrm{d}W(t)\,.$$

Here $g(t)$ is arbitrary smooth function. In the case of Shrödinger Bridge we take $g(t) = \sigma_e\sqrt{t(1-t)}$.

## E  Detail on the SDE case

### E.1  Optimal vector field and score for stochastic map

Following [20] we consider a so-called *Brownian bridge* $B(t)$ from $x_0$ to $x_1$ with constant diffusion rate $\sigma_e$. This stochastic process can be expressed through a multidimensional standard Winner process $W(t)$ as

$$B(t \mid x_0, x_1) = (1 - t)x_0 + tx_1 + \sigma_e(1 - t)W\left(\frac{t}{1 - t}\right). \tag{40}$$

Thus, the conditional distribution $p(t, x \mid x_0, x_1)$ conditioned on the starting $x_0$ and end point $x_1$ is Gaussian:

$$p(x, t \mid x_0, x_1) = \mathcal{N}\left(x \mid (1 - t)x_0 + tx_1, \sigma_e^2 t(1 - t)\right).$$

We can not directly use the results Theorem 3 from [9] (or similar Theorem 2.1 from [19] ) for the Gaussian paths, as in this case $\sigma(0) = 0$. To circumvent this obstacle and to be able to write an expression for the conditional velocity, we assume that we have a Gaussian distribution with a very narrow peak at the initial ($t = 0$) and final ($t = 1$) points. In other words, we will consider conditional probabilities of the form

$$p(x, t \mid x_0, x_1) = \mathcal{N}\left(x \mid (1 - t)x_0 + tx_1, \sigma_e^2(t + \eta)(1 - t + \eta)\right), \tag{41}$$

where parameter $\eta$ is small enough. Then we can use the above Theorems and immediately write

$$v_{x_0, x_1}(x, t) = \frac{\sigma'(t)}{\sigma(t)}\left(x - \mu(t)\right) + \mu'(t) = \frac{1 - 2t}{2(t + \eta)(1 - t + \eta)}\left(x - (1 - t)x_0 - tx_1\right) + x_1 - x_0. \tag{42}$$

After integrating over $x_0$ and $x_1$, we can take the limit $\eta \to 0$. Thus, now for fixed $x_0$ and $x_1$ we do not have a fixed value of $x_t$ in which to train the model, but a random one. In general case, we end up to the loss:

$$\mathcal{L}_v = \mathbb{E}_{t \sim \mathcal{U}(0,1),\, \{x_1, x_0\} \sim \pi,\, x \sim p(\cdot, t \mid x_0, x_1)}\|v_\theta(x, t) - v_{x_0, x_1}(x, t)\|^2, \tag{43}$$

where $\pi(x_1, x_0)$ is the density of the joint distributions with the marginal equal to the two given probabilities:

$$\int \pi(x_1, x_0)\, \mathrm{d}x_1 = \rho_0(x_0), \quad \int \pi(x_1, x_0)\, \mathrm{d}x_0 = \rho_1(x_1).$$

In the simple case, $\pi(x_1, x_0) = \rho_0(x_0)\rho_1(x_1)$. Vector field in Eq. (43) if taken in the form of Eq. (42).

Now, we can obtain an explicit form for the vector field $v$ at which the written loss is reached its minimum by performing the same calculations as in the derivation of formula (10):

$$v(x, t) = \frac{\iint v_{x_0, x_1}(x, t)\, p(x, t \mid x_0, x_1)\, \pi(x_0, x_1)\, \mathrm{d}x_0\, \mathrm{d}x_1}{\iint p(x, t \mid x_0, x_1)\, \pi(x_0, x_1)\, \mathrm{d}x_0\, \mathrm{d}x_1}. \tag{44}$$

As in the work [20] we can also train score network. Namely, as marginals for Brownian bridge are Gaussian, we can write explicit conditional score for conditional probabilistic path

$$\nabla \log p(x, t \mid x_0, x_1) = \frac{\mu(t) - x}{\sigma_e^2(t)} = \frac{x_0(1 - t) + x_1 t - x}{\sigma_e^2 t(1 - t)}.$$

In the work [20] the following loss is introduced to train a model for this score

$$\mathcal{L}_s = \mathbb{E}_{t \sim \mathcal{U}(0,1),\, \{x_1, x_0\} \sim \pi,\, x \sim p(\cdot, t \mid x_0, x_1)}\|s_\theta(x, t) - \nabla \log p(x, t \mid x_0, x_1)\|^2. \tag{45}$$

Similar to (44), for the optimal score $s$ we have:

$$s(x, t) = \frac{\iint \nabla \log p(x, t \mid x_0, x_1)\, p(x, t \mid x_0, x_1)\, \pi(x_0, x_1)\, \mathrm{d}x_0\, \mathrm{d}x_1}{\iint p(x, t \mid x_0, x_1)\, \pi(x_0, x_1)\, \mathrm{d}x_0\, \mathrm{d}x_1}, \tag{46}$$

where $p$ is given in (41).

## E.2 Use stochastic

Note that the obtained vector field gives marginal distributions $p(x, t)$, which (in the limit $\eta \to 0$) at $t = 1$ leads to the distribution we need: $p(x, t = 1) = \rho_1(x)$. However, the addition of the stochastic term allows us to extend the scope of application of the explicit formula for the vector field. In particular, it can be applied to the situation when we have two sets of samples and both distributions are unknown, as well as the possibility of constructing SDE and solving it using, for example, the Euler–Maruyama method (see examples below).

As consequence of Theorem 3.1 from [20] we have that, if $v$ is given by Eq. (44) then ODE

$$\frac{\partial \rho(x, t)}{\partial t} = -\operatorname{div}\big(\rho(x, t)v(x, t)\big) \tag{47}$$

recovers the marginal $\rho(x, t)$ (with the given initial conditions) of the stochastic process $P(t)$ which is obtained by marginalization conditional Brownian bridge (40) over initial and target distribution

$$P(t) = \int B(t \mid x_0, x_1)\pi(x_0, x_1)\,\mathrm{d}x_0\,\mathrm{d}x_1\,.$$

As the second consequence of this Theorem, the SDE

$$\mathrm{d}x(t) = \left(v\big(x(t), t\big) + \frac{g^2(t)}{2}s\big(x(t), t\big)\right)\mathrm{d}t + g(t)\,\mathrm{d}W(t) \tag{48}$$

generates so-called Markovization of the process $P(t)$. Indeed, we can rewrite PDE Eq. (47) in the form

$$\frac{\partial \rho(x, t)}{\partial t} = -\operatorname{div}\left(\rho(x, t)v(x, t) + \frac{g^2(t)}{2}\boldsymbol{\nabla}\rho(x, t)\right) + \frac{g^2(t)}{2}\Delta\rho(x, t),$$

where nabla operator is defined as $\Delta = \operatorname{div}\boldsymbol{\nabla}$. Thus, we get the Fokker–Planck equation for the density of the stochastic process (48).

## E.3 Particular cases

In particular case of Brownian bridge when $\sigma_e(t) = \sigma_\epsilon\sqrt{t(1-t)}$, then $\sigma_e'(t) = \sigma_\epsilon(1 - 2t)/\big(2\sqrt{t(1-t)}\big)$. In this section we consider simple case of separable variables $\pi(x_0, x_1) = \rho_0(x_0)\rho_1(x_1)$.

### E.3.1 Gaussian initial distribution

In the case, when $\rho_0$ is standard Gaussian distribution: $\rho_0 = \mathcal{N}\left(\cdot\,\middle|\,0, 1^2\right)$, we can take integral on $x_0$ and then take the limit $\eta \to 0$ in the expressions for $v$ and $s$. First, consider the expression for $v$: where we use explicit expression (41) for conditional density path and Eq. (42) for conditional velocity:

$$v(x, t) = \frac{\int w(x, t \mid x_1)\mathcal{N}\left(x\,\middle|\,x_1 t, \sigma_e^2 t(1-t) + (1-t)^2\right)\rho_1(x_1)\mathrm{d}x_1}{\int \mathcal{N}\left(x\,\middle|\,x_1 t, \sigma_e^2 t(1-t) + (1-t)^2\right)\rho_1(x_1)\mathrm{d}x_1} =$$

$$= \frac{\int w(x, t \mid x_1)\rho_0\left(\frac{x - x_1 t}{\sqrt{\sigma_e^2 t(1-t) + (1-t)^2}}\right)\rho_1(x_1)\mathrm{d}x_1}{\int \rho_0\left(\frac{x - x_1 t}{\sqrt{\sigma_e^2 t(1-t) + (1-t)^2}}\right)\rho_1(x_1)\mathrm{d}x_1}, \tag{49}$$

where $w(x, t \mid x_1)$ is the conditional velocity, generated by the conditional map $\phi_{t, x_1}(x) = \sqrt{\sigma_e^2 t(1-t) + (1-t)^2} + tx_1$:

$$w(x, t \mid x_1) = \frac{x_1 - x}{1 - t + t\sigma_e^2} + \sigma_e^2\frac{(1 - 2t)x + tx_1}{2\big((1-t)^2 + (1-t)t\sigma_e^2\big)}.$$

Thus, note that in the case of Gaussian distributions, all the difference between this expression and the expression without the stochastic part is the appearance of additional (time-dependent, in general) variance. Marginal distributions are still Gaussian's.

878 Similar, using Eq. (46) we have for the score $s$:

$$s(x,t) = \frac{\int (tx_1 - x)\mathcal{N}\left(x\,|\,x_1 t, \sigma_e^2 t(1-t) + (1-t)^2\right)\rho_1(x_1)\mathrm{d}x_1}{\left((1-t)^2 + (1-t)t\sigma_e^2\right)\int \mathcal{N}\left(x\,|\,x_1 t, \sigma_e^2 t(1-t) + (1-t)^2\right)\rho_1(x_1)\mathrm{d}x_1} =$$

$$= \frac{\int (tx_1 - x)\rho_0\left(\frac{x - x_1 t}{\sqrt{\sigma_e^2 t(1-t) + (1-t)^2}}\right)\rho_1(x_1)\mathrm{d}x_1}{\left((1-t)^2 + (1-t)t\sigma_e^2\right)\int \rho_0\left(\frac{x - x_1 t}{\sqrt{\sigma_e^2 t(1-t) + (1-t)^2}}\right)\rho_1(x_1)\mathrm{d}x_1}. \quad (50)$$

### 879 E.3.2 Samples instead of distributions

880 Consider the case where we only have access to the samples $\{x_0^i\}_{i=1}^{N_0}$ and $\{x_1^i\}_{i=1}^{N_1}$ from both
881 distributions, $\rho_0$ and $\rho_1$, but do not know their explicit expressions. In this case, we can estimate the
882 vector field using by a method similar to the one we used to estimate the vector field in (15):

$$v(x,t) \approx \frac{\sum_{i=1}^{N_0} \sum_{j=1}^{N_1} v_{x_0^i, x_1^j}(x,t)\, p(x,t \mid x_0^i,\, x_1^j)}{\sum_{i=1}^{N_0} \sum_{j=1}^{N_1} p(x,t \mid x_0^i,\, x_1^j)}. \quad (51)$$

883 Similar for the score

$$s(x,t) \approx \frac{\sum_{i=1}^{N_0} \sum_{j=1}^{N_1} \boldsymbol{\nabla} p(x,t \mid x_0^i, x_1^j)\, p(x,t \mid x_0^i,\, x_1^j)}{\sum_{i=1}^{N_0} \sum_{j=1}^{N_1} p(x,t \mid x_0^i,\, x_1^j)}. \quad (52)$$

884 In addition, we can also use the importance sampling method in this case. Namely we can use
885 both approaches: self-normalized importance sampling and rejection sampling, similar to what is
886 described in Sec. B

## 887 F Consistency of Eq. (24) in the case of optimal transport

Let us analyze what happens if in formula (24) the joint density $\pi$ represents the following Dirac
delta-function[4]:

$$\pi(x_0, x_1) = \delta\big(x_0 - F(x_1)\big),$$

888 *i. e.* we have a deterministic mapping $F$ from $x_1$ to $x_0$. Then, the Eq. (34) come to

$$v(x,t) = \frac{\int (x_1 - x)\,\delta\big(\phi_{t,x_1}^{-1}(x) - F(x_1)\big)\,\mathrm{d}x_1}{(1-t)\int \delta\big(\phi_{t,x_1}^{-1}(x) - F(x_1)\big)\,\mathrm{d}x_1}.$$

889 Let $y(x,t)$ be the unique solution of the equation

$$\phi_{t,y}^{-1}(x) = F(y), \quad (53)$$

considered as an equation on $y$. Then

$$v(x,t) = \frac{x - y(x,t)}{1-t}.$$

Now, let us use linear mapping $\phi_{t,x_1}(x) = x_1 t + x(1-t)$, with inverse $\phi_{t,x_1}^{-1}(x) = \frac{x - tx_1}{1-t}$, and
consider the simplest case when the original distribution is a $d$-dimensional standard Gaussian and $\rho_1$
is a $d$-dimensional Gaussian with mean $\mu$ and diagonal variance $\Sigma = \mathrm{diag}(\sigma)$. We know the OT
correspondence between Gaussians, namely

$$\big(F(x_1)\big)_i = \frac{(x_1 - \mu)_i}{\Sigma_{ii}}, \quad \forall 1 \geq i \geq d.$$

---

[4]Further reasoning is not absolutely rigorous, and in order not to introduce the axiomatics of generalized
functions, we can assume that the delta function is the limit of the density of a normal distribution with mean 0
and variance tending to zero.

Here and further by index $i$ we denote $i$th component of the corresponding vector. Then, the Eq. (53) reads as

$$\frac{(x - yt)_i}{1 - t} = \frac{(y - \mu)_i}{\Sigma_{ii}},$$

with the solution

$$\big(y(x, t)\big)_i = \frac{\mu_i(1 - t) + x_i\Sigma_{ii}}{1 + (\Sigma_{ii} - 1)t}.$$

Then the expression for the vector field is

$$\big(v(x, t)\big)_i = \frac{\mu_i + x_i(\Sigma_{ii} - 1)}{1 + (\Sigma_{ii} - 1)t}.$$

Now, knowing the expression for velocity, we can write the equations for the trajectories $x(t)$:

$$\begin{cases} \big(x'(t)\big)_i = \dfrac{\mu_i + (x(t))_i(\Sigma_{ii} - 1)}{1 + (\Sigma_{ii} - 1)t}, \\ x(0)_i = (x_0)_i \end{cases}.$$

This equation have closed-form solution:

$$x(t) = \mu t + x_0 - (1 - \sigma)\, tx_0.$$

Analyzing the obtained solution, we conclude that, first, the trajectories obey the given mapping $F$:

$$\big(F(x(1))\big)_i = (x_0)_i = \frac{(x(1) - \mu)_i}{\Sigma_{ii}},$$

And, second, the trajectories are straight lines (in space), as they should be when the flow carries points along the optimal transport.

As a final conclusion, note that, of course, if we are mapping optimal transport $F$, then it is meaningless to use numerical formula (16). However, usually the exact value of the mapping $F$ is not known, and our theoretical formula (34) can help to rigorously establish the error that is committed when an approximate mapping is used instead of the optimal one.

# G  Analytical derivations for example in Fig. 1(b)

## G.1  CFM dispersion

To derive the analytical expression for the optimal flow velocity in the case of two normal distributions $\rho_0 \sim N(0, I)$ and $\rho_1 \sim N(\mu, \sigma^2 I)$, we start by substituting $\mu_0 = 0$, $\sigma_0 = 1$, $\mu_1 = \mu$, $\sigma_1 = \sigma$, to the exact expression (35) to get

$$v(x, t) = \frac{t\sigma^2 + t - 1}{(1 - t)^2 + t^2\sigma^2}x + \frac{1}{(1 - t)^2 + t^2\sigma^2}(\mu - t\mu) = w(t)x + C, \tag{54}$$

where

$$w(t) = \frac{t\sigma^2 + t - 1}{(1 - t)^2 + t^2\sigma^2},$$

and $C$ is constant independent of $x$. We then redefine the dispersion based on Eq. (19) using $x = (1 - t)x_0 + tx_1$ with $x_0 \sim \rho_0$ and $x_1 \sim \rho_1$:

$$\mathbb{D}_{x, x_1} f(x,\, x_1) = \mathbb{D}_{x_0, x_1} f\big((1 - t)x_0 + tx_1,\, x_1\big) \tag{55}$$

This leads us to the final expression:

$$\mathbb{D}_{x, x_1}\Delta v(x, t) = \mathbb{D}_{x_0, x_1}((1 - w(t))x_1 - (1 + w(t)(1 - t))x_0) =$$
$$= (1 + w(t)(1 - t))^2\mathbb{D}_{x_0}x_0 + (1 - w(t))^2\mathbb{D}_{x_1}x_1.$$

This provides a comprehensive representation of the updated dispersion for the CFM objective at any given time $t$.

---

**Algorithm 2** Computation ExFM dispersion algorithm

---

**Require:** Density function for initial distribution $\rho_0$; sampler for target distribution $\rho_1$; parameter $M$ (number of samples for evaluation); parameter $N$ (number of samples from $\rho_1$ for certain samples $x \sim \rho_m(x,t)$); optimal model $v(x,t)$; time for evaluation $t$.

**Ensure:** numerical evaluation of dispersion update for ExFM objective

1: Sample $(M \cdot N)$ samples $x_1^{i,j}$ from $\rho_1$, where $i \in [1, M]$ and $j \in [1, N]$
2: Sample $(M)$ samples $x_0^i$ from $\rho_0$, where $i \in [1, M]$
3: Compute points $x^i$ as $(1-t)x_0^i + tx_1^{i,0}$
4: Compute $v^d(x^i, t) = \sum_{j=1}^{N} \tilde{\rho}^{i,j}(t)\frac{x_1^{i,j}-x^i}{1-t}$, where $\tilde{\rho}^{i,j}(t) = \rho_0\left(\frac{x^i - tx_1^{i,j}}{1-t}\right) / \sum_{j=1}^{N} \rho_0\left(\frac{x^i - tx_1^{i,j}}{1-t}\right)$
5: Compute and return dispersion $\mathbb{D}_i(v(x^i, t) - v^d(x^i, t))$

---

## G.2   ExFM dispersion

The analytical derivation of the updated dispersion for the ExFM objective proves to be complex in practice. Therefore, for the example at hand, a numerical scheme was employed for evaluation. The procedure outlined in Alg. 2 was utilized for this task. The experiment's parameters for the algorithm were as follows: $M = 200k$, $N = 128$, $\rho_0 = N(0, I)$, $\rho_1 = N(\mu, \sigma^2 I)$, and the optimal model $v(x,t)$ was derived from equation (54).

# H   Additional Experiments

## H.1   2D toy examples

To ensure the reliability and impartiality of the outcomes, we carried out the experiment under uniform conditions and parameters. Initially, we generated a training set of batch size $N = 10,000$ points. The employed model was a simple Multilayer Perceptron with ReLu activations and 2 hidden layers of 512 neurons, `Adam` optimizer with a learning rate of $10^{-3}$, and no learning rate scheduler. We determined the number of iteration steps equal to 10000. Subsequently, we configured the mini batch size $n = 256$ during the training procedure, with the primary objective of minimizing the Mean Squared Error (MSE) loss. The full training algorithm and notations can be seen in Algorithm 1. To perform sampling, we employed the function `odeint` with `dopri5` method from the python package `torchdiffeq import odeint` with `atol` and `rtol` equal $1e-5$.

## H.2   Tabular

The `power` dataset (dimension = 6, train size = 1659917, test size = 204928) consisted of electric power consumption data from households over a period of 47 months. The `gas` dataset (dimension = 8, train size = 852174, test size = 105206) recorded readings from 16 chemical sensors exposed to gas mixtures. The `hepmass` dataset (dimension = 21, train size = 315123, test size = 174987) described Monte Carlo simulations for high energy physics experiments. The `minibone` (dimension = 43, train size = 29556, test size = 3648) dataset contained examples of electron neutrino and muon neutrino. Furthermore, we utilized the `BSDS300` dataset (dimension = 63, train size = 1000000, test size = 250000), which involved extracting random 8 x 8 monochrome patches from the BSDS300 datasets of natural images [11].

These diverse multivariate datasets are selected to provide a comprehensive evaluation of performance across various domains. To maintain consistency, we followed the code available at the given GitHub link[5] to ensure that the same instances and covariates were used for all the datasets.

To ensure the correctness of the experiments we conduct them with the same parameters. To train the model we use the same MultiLayer Perceptron (1024 x 3) model with ReLu activations, `Adam` as optimizer with learning rate of $10^{-3}$ and no learning rate scheduler. As in the pretrained step, we use separately training and testing sets for training the model and calculating metrics. We train the models on the full dataset (of size `train_set_size`) with batch size $N = 5000$ (`batch_size`)

---

[5]https://github.com/gpapamak/maf

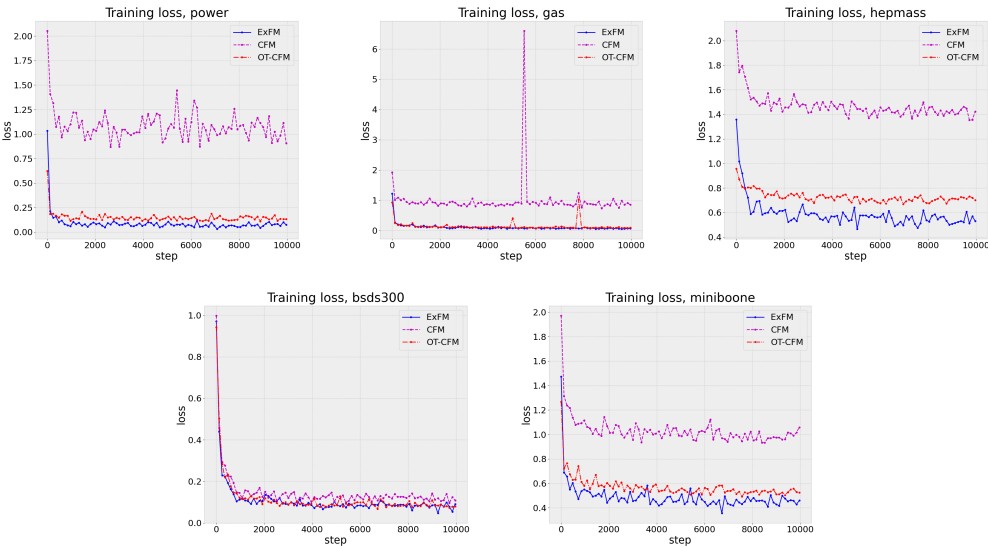

Figure 7: Training loss comparison for ExFM, CFM and OT-CFM methods over $10\,000$ learning steps.

(except `miniboone` dataset, here we used 2000 since the smaller size of the dataset) and mini batches $n = 256$ elements (`mini_batch_size`), the number of epochs and steps for each dataset is adaptive `num_epochs = train_set_size // batch_size` and `num_steps = batch_size // mini_batch_size`.

For both 2D-toy an tabular data: we take $m = n$ time variable, individual value of variable $t$ corresponds to its pair $(x_0, x_1)$. The notations $N$, $n$ and $m$ corresponds to those in Algorithm 1. To perform sampling, we employed the function `odeint` with `dopri5` method from the python package `torchdiffeq import odeint` with `atol` and `rtol` equal $1e - 5$.

Table 5: Learning parameters for Tabular datasets.

| DATA | MLP LAYERS | LR |
|------|------------|-----|
| POWER | [512, 1024, 2048] | 1E-3 |
| GAS | [512, 1024,1024] | 1E-4 |
| HEPMASS | [512, 1024] | 1E-3 |
| BSDS300 | [512, 1024,1024] | 1E-4 |
| MINIBOONE | [512, 1024] | 1E-3 |

Table 6: NLL comparison for ExFM, CFM and OT-CFM methods over $10\,000$ learning steps, mean and std taken from 10 sampling iterations.

| DATA | ExFM | CFM | OT-CFM |
|------|------|-----|--------|
| POWER | **-8.51e-02 $\pm$ 4.85e-02** | 1.64E-01 $\pm$ 4.18E-02 | 5.22E-02 $\pm$ 3.92E-02 |
| GAS | **-5.53e+00 $\pm$ 3.66e-02** | -5.00E+00 $\pm$ 2.56E-02 | -5.48E+00 $\pm$ 2.90E-02 |
| HEPMASS | 2.16E+01 $\pm$ 6.31E-02 | **2.21e+01 $\pm$ 6.13e-02** | 2.16E+01 $\pm$ 4.32E-02 |
| BSDS300 | -1.29E+02 $\pm$ 8.40E-01 | -1.29E+02 $\pm$ 8.97E-01 | **-1.32e+02 $\pm$ 6.39e-01** |
| MINIBOONE | **1.34e+01 $\pm$ 1.95e-04** | 1.42E+01 $\pm$ 1.29E-04 | 1.43E+01 $\pm$ 9.22E-05 |

### H.3 ExFM-S evaluation

The models were assessed using four toy datasets of two dimensions each. A three-layer MLP network was utilized, featuring SeLU activations and a hidden dimension of 64. Optimization was carried out using the AdamW optimizer with a learning rate of $10^{-3}$ and a weight decay of $10^{-5}$.

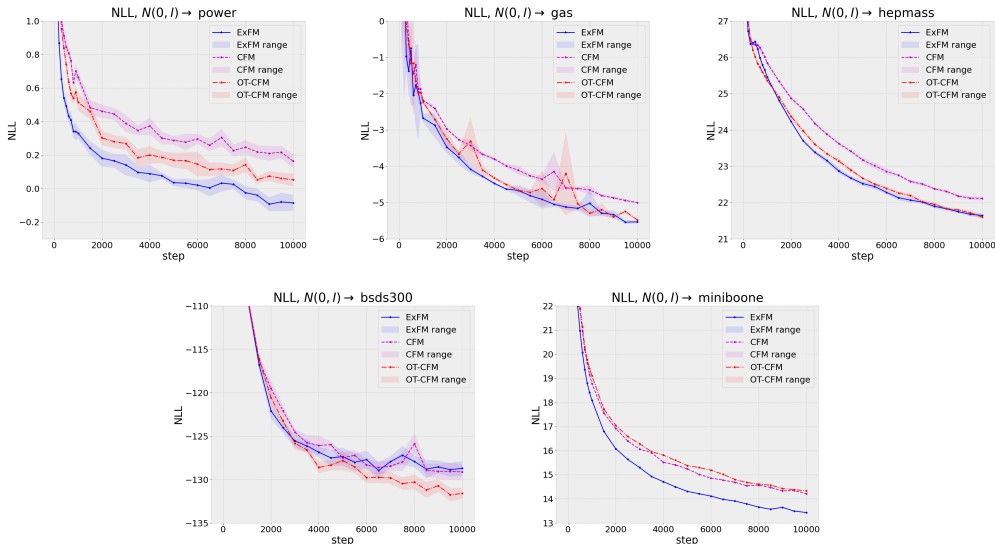

Figure 8: NLL comparison for ExFM, CFM and OT-CFM methods over 10 000 learning steps, mean and std for range taken from 10 sampling iterations.

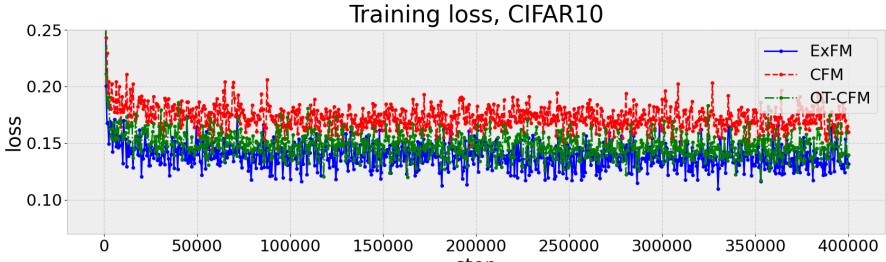

Figure 9: Training loss comparison for ExFM, CFM and OT-CFM methods, CIFAR-10 dataset.

The model was trained over 2,000 iterations with a batch size of 128. Inference was conducted using the Euler solver for Ordinary Differential Equations (ODE) with 100 steps. To validate the models, the POT library was employed to compute the Wasserstein distance based on 4,000 samples. The experiments were performed on a single Nvidia H100 GPU with 80gb memory.

## H.4 CIFAR 10 and MNIST

We conducted experiments related to high dimensional data, the parameters for training were taken from the open-source code[6] from the works [20, 19]. We saved the leverage of additional heuristics(EMA, lr scheduler).

Table 7: FID comparison for 4 sampling iterations, 400 000 learning steps.

| METHOD | FID |
|--------|-----|
| EXFM | $\mathbf{3.686 \pm 0.029}$ |
| CFM | $3.727 \pm 0.026$ |
| OT-CFM | $3.843 \pm 0.033$ |

---

[6]https://github.com/atong01/conditional-flow-matching

Table 8: FID comparison for ExFM, CFM and OT-CFM methods over 400 000 learning steps, mean and std taken from 4 sampling iterations.

| Step | ExFM FID | CFM FID | OT-CFM FID |
|---|---|---|---|
| 0 | $447.256 \pm 0.116$ | $447.106 \pm 0.130$ | $447.091 \pm 0.081$ |
| 20000 | $281.060 \pm 0.243$ | $275.044 \pm 0.123$ | $281.499 \pm 0.287$ |
| 40000 | $52.050 \pm 0.245$ | $51.436 \pm 0.142$ | $45.976 \pm 0.109$ |
| 60000 | $\mathbf{9.125 \pm 0.060}$ | $9.181 \pm 0.035$ | $10.358 \pm 0.054$ |
| 80000 | $\mathbf{6.624 \pm 0.053}$ | $6.978 \pm 0.062$ | $7.492 \pm 0.050$ |
| 100000 | $\mathbf{5.641 \pm 0.048}$ | $5.894 \pm 0.045$ | $6.299 \pm 0.031$ |
| 120000 | $\mathbf{5.085 \pm 0.031}$ | $5.247 \pm 0.051$ | $5.558 \pm 0.017$ |
| 140000 | $\mathbf{4.766 \pm 0.036}$ | $4.902 \pm 0.053$ | $5.120 \pm 0.043$ |
| 160000 | $\mathbf{4.486 \pm 0.054}$ | $4.593 \pm 0.068$ | $4.828 \pm 0.046$ |
| 180000 | $\mathbf{4.294 \pm 0.023}$ | $4.447 \pm 0.045$ | $4.576 \pm 0.051$ |
| 200000 | $\mathbf{4.180 \pm 0.029}$ | $4.204 \pm 0.013$ | $4.434 \pm 0.031$ |
| 220000 | $\mathbf{4.022 \pm 0.036}$ | $4.182 \pm 0.024$ | $4.331 \pm 0.036$ |
| 240000 | $\mathbf{3.925 \pm 0.028}$ | $4.037 \pm 0.036$ | $4.227 \pm 0.050$ |
| 260000 | $\mathbf{3.852 \pm 0.047}$ | $3.937 \pm 0.018$ | $4.125 \pm 0.015$ |
| 280000 | $\mathbf{3.842 \pm 0.053}$ | $3.870 \pm 0.040$ | $4.056 \pm 0.029$ |
| 300000 | $\mathbf{3.758 \pm 0.032}$ | $3.788 \pm 0.024$ | $4.017 \pm 0.029$ |
| 320000 | $\mathbf{3.749 \pm 0.029}$ | $3.792 \pm 0.034$ | $3.937 \pm 0.052$ |
| 340000 | $\mathbf{3.724 \pm 0.042}$ | $3.747 \pm 0.033$ | $3.897 \pm 0.037$ |
| 360000 | $\mathbf{3.714 \pm 0.022}$ | $3.751 \pm 0.041$ | $3.875 \pm 0.015$ |
| 380000 | $\mathbf{3.707 \pm 0.028}$ | $3.754 \pm 0.020$ | $3.917 \pm 0.037$ |
| 400000 | $\mathbf{3.686 \pm 0.029}$ | $3.727 \pm 0.026$ | $3.843 \pm 0.033$ |

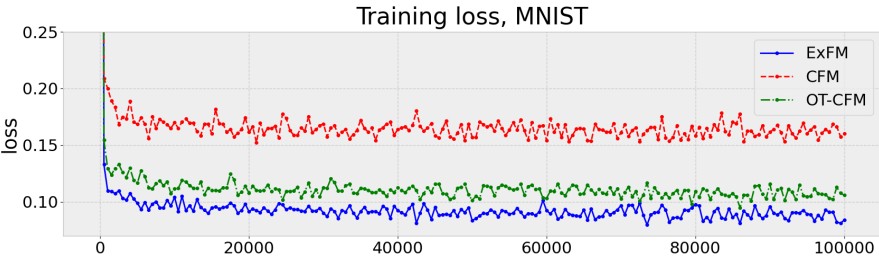

Figure 10: Training loss comparison for ExFM, CFM and OT-CFM methods, MNIST dataset.

### H.5 Metrics

For evaluating 2D toy data we use Energy Distance and W2 metricis, for Tabular datasets we use Negative Log Likelihood, for CIFAR10 we took Fréchet inception distance (FID) metrics. This choice is connected with an instability and poor evaluation quality of Energy Distance metrics and W2 among high-dimensional data .

### H.5.1 Energy Distance

We use the generalized Energy Distance [18] (or E-metrics) to the metric space.

Consider the null hypothesis that two random variables, $X$ and $Y$, have the same probability distributions: $\mu = \nu$ .

For statistical samples from $X$ and $Y$:

$$\{x_1, \ldots, x_n\} \quad \text{and} \quad \{y_1, \ldots, y_m\},$$

the following arithmetic averages of distances are computed between the $X$ and the $Y$ samples:

$$A = \frac{1}{nm} \sum_{i=1}^{n} \sum_{j=1}^{m} \|x_i - y_j\|, \quad B = \frac{1}{n^2} \sum_{i=1}^{n} \sum_{j=1}^{n} \|x_i - x_j\|, \quad C = \frac{1}{m^2} \sum_{i=1}^{m} \sum_{j=1}^{m} \|y_i - y_j\|.$$

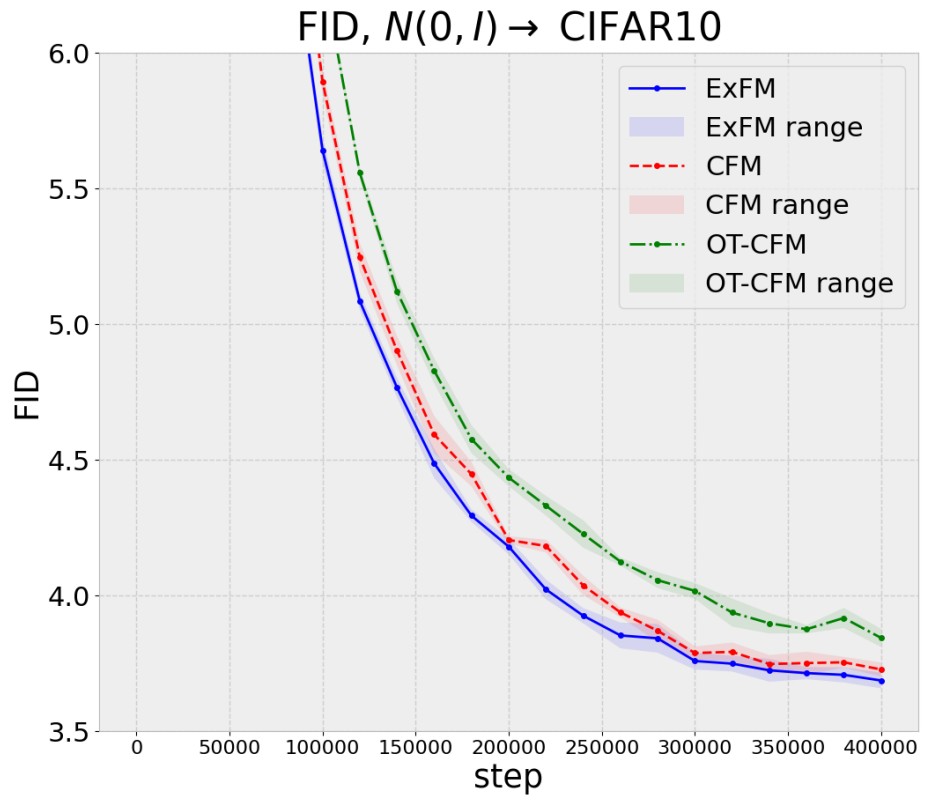

Figure 11: FID comparison for ExFM, CFM and OT-CFM methods, CIFAR-10 dataset.

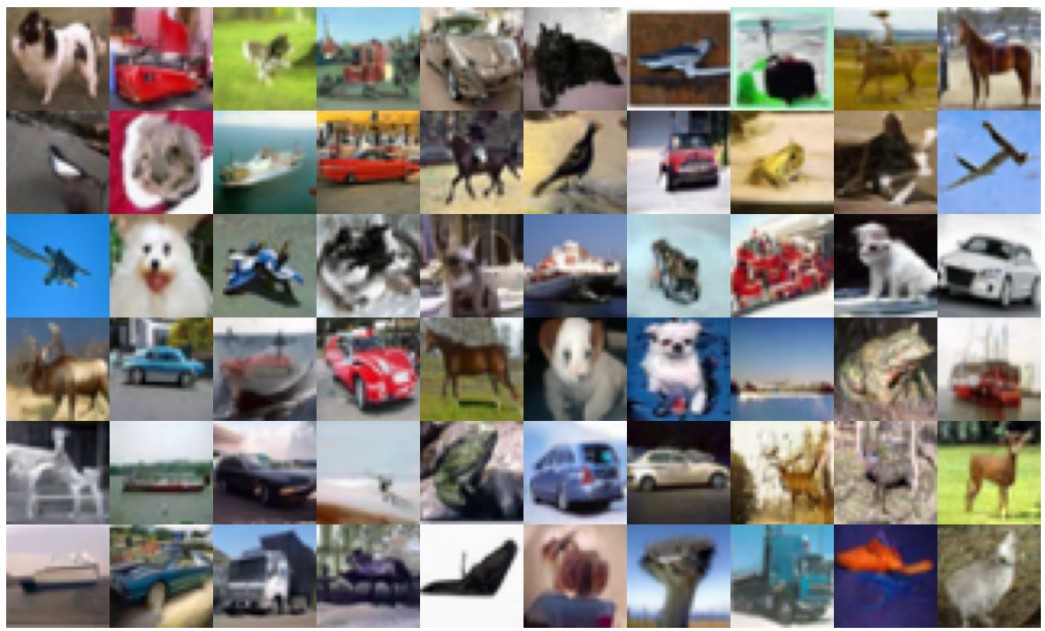

Figure 12: Sampled images from ExFM method, CIFAR-10 dataset.

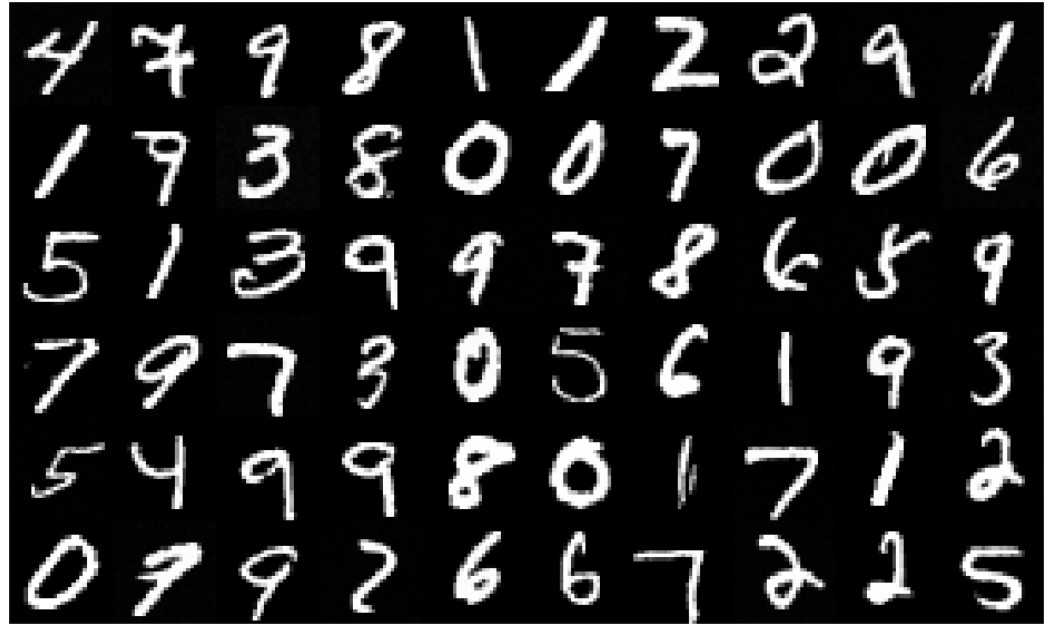

Figure 13: Sampled images from ExFM method, MNIST dataset.

The E-statistic of the underlying null hypothesis is defined as follows:

$$E_{n,m}(X, Y) := 2A - B - C$$

### H.5.2 2-Wasserstein distance (W2)

The 2-Wasserstein distance [14], also called the Earth mover's distance or the optimal transport distance $W$ is a metric to describe the distance between two distributions, representing two different subsets $A$ and $B$. For continuous distributions, it is:

$$W := W(F_A, F_B) = \left( \int_0^1 \left| F_A^{-1}(u) - F_B^{-1}(u) \right|^2 du \right)^{\frac{1}{2}},$$

where $F_A$ and $F_B$ are the corresponding cumulative distribution functions and $F_A^{-1}$ and $F_B^{-1}$ the respective quantile functions.

### H.5.3 Negative Log Likelihood (NLL)

To compute the NLL, we first sampled $N = 5000$ samples $\{x_i^s\}_{i=1}^N$ from the target distribution. Then we solved the following inverse flow ODE:

$$\begin{cases} \dfrac{\partial x(t)}{\partial t} = v_\theta(x(t), t), \\ x(1) = x_s \end{cases}$$

for $t$ from 1 to 0. For simplicity, changing time variable $\tau = 1 - t$ we solve the following ODE:

$$\begin{cases} \dfrac{\partial x(\tau)}{\partial \tau} = -v_\theta(x(\tau), 1 - \tau), \\ x(0) = x_s \end{cases}$$

for $\tau$ from 0 to 1. Thus we obtained $N$ solutions $\{x_i^0\}_{i=1}^N$ which are expected to be distributed according to the standard normal distribution $\mathcal{N}(x \mid 0, I)$. So we calculate NLL as

$$\text{NLL} = -\frac{1}{N} \sum_{i=1}^N \ln \mathcal{N}(x_i^0 \mid 0, I).$$

### H.5.4  Fréchet inception distance (FID)

For images evaluation we take Fréchet inception distance (FID) metrics, in particular the implementation from [12]. The main idea of FID metrics is to measure the gap between two data distributions, such as between a training set and samples from a trained model. After resizing the images, and feature extraction, the mean $(\mu, \hat{\mu})$ and covariance matrix $(\Sigma, \hat{\Sigma})$ of the corresponding features are used to compute FID:

$$\text{FID} = ||\mu - \hat{\mu}||_2^2 + \text{Tr}(\Sigma + \widehat{\Sigma} - 2(\Sigma\hat{\Sigma})^{1/2}),$$

where $Tr$ is the trace of the matrix.

