# OpenReview forum: "Explicit Flow Matching: On The Theory of Flow Matching Algorithms with Applications"
_NeurIPS.cc/2024/Conference — Submitted to NeurIPS 2024_

### Official Review · Reviewer_kwRu · 2024-06-14

**Soundness:** 2
**Presentation:** 2
**Contribution:** 3
**Rating:** 5
**Confidence:** 4

**Summary:**

The paper proposes a loss for training flow matching (rectified flows, stochastic interoplants) that is based on integrating the target velocity over the available data as the regression target. This reduces the variance of the gradient estimator and can also be applied in the stochastic variant of flow matching.

**Strengths:**

To the best of my knowledge, the idea is novel to use a larger batch of data for the velocity target in the flow matching objective than as the input for the network. This reduces the variance in the loss estimate.

The additional compute over the FM loss seems negligible, since only the target velocity field is adapted, but the network still receives batches. This makes it an attractive improvement to flow matching training.

**Weaknesses:**

I genuinely like the idea, but cannot recommend acceptance in the current presentation. I am happy to be corrected on any point and adjust my score.

Below I grouped the weaknesses by category.

## Theoretical contributions can be simplified and potentially reveal existing result (update: largely addressed, but presentation to be improved)

I think the theoretical derivation can be greatly simplified:

1. The notation is overly complicated. Why not use the standard notation for joint $p(a, b)$, conditional $p(a|b)$ and marginal $p(a)$ probabilities instead of index notation, where the index sometimes means "joint", "conditional" or "marginal", and sometimes indexes time $t$ or a condition such as $x_1$? (or $\\rho$ instead of $p$).
2. I think this reveals that the new loss is simply obtained by writing the target velocity in the ExFM loss as the expectation over the training data $\\rho_1$: Eqs (8, 10) say that the target velocity is given by $\\int w(t, x_1, x) \\rho(x|x_1, t) dx $, that is just average the velocities over the entire training data, weighted by the probability that the path actually  (where $x$ is sampled from the linear conditional paths). Given this observation, it seems to me that the actual contribution of the paper does not lie in this new loss, but how to efficiently estimate this integral, which is currently hidden in Appendix B.

In fact, I think that sections 2.1 and 2.2 can be merged to a simple importance sampling argument in the original flow matching argument.

I also think that the authors are missing that their third contribution has already been derived in the same form by their reference [10] in Eq. 4 and I think the second contribution is a simple extension to different conditional flow fields resulting from different inversions $\\varphi^{-1}$.

## Evaluation on tabular data is wrong (update: fixed)

The NLL defined in Appendix H.5.3 does not contain the volume change, which is an integral part of the negative log-likelihood. Did you use this equation for evaluation? My current score reflects the belief that volume change was accounted for.

Also, in Table 3, sometimes the highest values and sometimes the lowest values in each row are marked bold. Which model is better and does this use the incorrect formula? Please update without e-notation, adapting -1.29E+02 to -129, this is hard to read.

## Toy data evaluation (update: fixed)

It is easy to construct a very good approximation for the moons distribution by taking a Gaussian mixture of values for Table 4.

## Typos (has no influence on my recommendation)

Here is a list of what I found:

- l. 30: introduced -> introduce
- l. 33: base -> based
- l. 65: $rho$ -> $\\rho$
- l. 70: need -> needed
- l. 92/93: using map -> using the map
- l. 100: we return to end of the standard CFM loss representation -> ?
- l. 105: just (unknown) -> just the (unknown)
- eq. 7: the integral shares variables with the outside expression, e.g. add tilde on the integration variables
- l. 124: inevitable -?> invertible
- l. 126: have -> has
- eq. 16: consider moving numerical tricks like using softmax from the theory section of the paper, page 6 already introduces a lot of notation.
- throughout: dispersion -> variance

**Questions:**

- Is the variance a problem in high dimensions? I would expect that overall there are few collisions of conditional flows paths.
- It is not fair to compare the MSE value of CFM and ExFM directly in Table 2, since they compute different quantities, so why list it?
- Why would the authors expect that their method performs beneficially in terms of the optimal transport distance (NPE)? OT-FM is explicitly built to reduce it, why would ExFM provide a benefit since it has the same minimum as CFM?
- What is the difference between Table 3 and 6?

**Limitations:**

I do not think that the limitations of the work are properly addressed in the theoretic part, in contrary to the statement of the authors in the paper checklist. In particular, I did not fully understand how accurately Eq. (10) (which is part of the loss) can be estimated. One question that can be a way towards addressing this is by expanding on when the assumption in line 172/173 is valid (and why this Jacobian is even a problem).

---

> ### Author Rebuttal · Authors · 2024-08-07
>
> Thank you for your detailed analysis of our work.
>
> **Let us discuss your comments:**
>
> * ``The notation is overly...``
>
>  We've changed the notations and listed changes in the 1-page PDF in the "all reviewers answer".
> Wherever we refer to the probability density function,we use the symbol $\rho$ (or similar), whose arguments are real numbers.
> Thus we emphasize that these are ordinary functions from the point of view of mathematical analysis; while the letter "P" of the form $\mathbb P$ is usually used to denote the probability of an event that is included as an argument.
> We have tried to maintain mathematical rigor, since our paper is positioned as a theoretical paper.
> As is customary in probability theory, the expression for the conditional probability $A | B$ used only in cases where $A$ and $B$ are understood as events or random variables (and not as values of random variables), and they are used only under symbols like $\mathbb P$, $\mathbb E$, i.e., when we are talking about the probability of an event, conditional expectation, etc.
>
> * ``This reveals that... ``
>
> We agree that the loss itself could be obtained more easily and with less computation.
> We have adressed this comment in the global answer.
>
> * ``The authors are missing that their third contribution ... ``
>
> We agree that this formula is somewhat analogous to ours, as other similar formulas have also can be found in literature.
> However, we respectfully disagree that these are consistent with our contribution.
> Indeed, our explicit form of loss immediately allows one to write the discrete loss as Eq. (13) or using other techniques from Appendix B to estimate the integral.
> To the best of our knowledge, such a training scheme has not been described in the literature before.
> In addition, our notation of loss allows, for example, one to obtain explicitly an expression for the vector field (Eq. (37)) in the case of a Gaussian initial distribution and Gaussian Mixture as the target distribution. Such an explicit expression for the velocity has not yet been described.
> Eq. (4) of [10] does not have such features.
> In addition, we also obtain formulas for the score like Eq. (46).
>
> Our second contribution says that we have obtained such a loss whose minimum can be equal to zero.
> In this case, the expression Eq. (4) from [10] as a small modification of the standard CFM loss does not possess this property.
>
> * ``The NLL defined...``
>
> We appreciate the reviewer's careful examination of our work. To compute the NLL, we follow Lipman et al. (2023), Appendix C, Eq. (27)–(33).
> So we calculate NLL as
> $\hbox{NLL}=-\frac1N \sum_{i=1}^N \Bigl(\ln \mathcal N(x^0_i \mid 0,I) + f^0_i \Bigr).$
>
> * `` In Table 3...``
>
> We thank the reviewer for pointing out the inconsistency in bolding within Table 3. We answered the comment in the global answer.
>
> * `` It is easy to ...``
>
> In the experiments in the Table 4 the primary experimental setup follows that used in [19].
> The main goal of these experiments was the proof-of-concept test of the performance of our velicity formula operating in the regime of unknown initial and target distributions against a fresh state-of-the-art approach.
>
> * `` I do not think that...``
>
> We provide some estimates of the error of computing the integral in Appendix B. However, we do not restrict the application of our algorithm to using Self-normalized Importance Sampling (SIS) to estimate the integral in Eq. (10), different methods (including the one described in Appendix B rejection sampling, or SIS with reduced bias) can be used, thus the accuracy of the estimation depends on the chosen method, and the choice of the best method and its error is a part of future work.
>
> The independence of the Jacobian from $x_1$ is checked directly for each chosen map.
> In all cases we used, this was true.
>
> **Answering your questions:**
>
> * `` Is the variance...``
>
> Indeed, as our experiments show, the presented algorithm performs much better on small dimensions.
> However, this property is also observed for other algorithms, say, the same is true for OT-CFM.
> Near the point $t=0$ our algorithm will still have a better variance as we have proven theoretically.
>
>  * `` It is not fair...``
>
> While we employed MSE as a loss function to demonstrate the impact of our proposed method's reduced variance on overall performance, a direct comparison between the two models can be misinterpreted. To address this, we have removed the aforementioned table as it lacked clarity. New tables can be seen in 1-page PDF.
>
> * `` Why would the authors...``
>
> We don't expect to be better than OT-CFM at an inference step. However, let us emphasize our key difference from the OT-CFM approach. The first is that we have certain theoretical guarantees, which we formulated in Theorem 2.4. and Proposition 2.5--2.6. OT-CFM uses minibatch-OT heuristics, for which, to the best of our knowledge, there is no such rigorous research.
>
> Second, our variance reduction algorithm is rather a by-product of our theoretical work.  And at the same time, our algorithm showed the same, or even better in some cases, variance reduction that the OT-CFM (specifically designed for this purpose) shows. In addition to the algorithm itself, we have derived a zoo of exact formulas in the paper, along with analysis of, for example, the case of stochastic additive, etc. OT-CFM, as we know, is only a heuristic aimed at obtaining a more efficient algorithm. On the contrary, we tried to present a theoretical analysis on the basis of which many future amendments to existing algorithms can be built or new ones can be invented.
> We hope that among these algorithms, there will be some that will reduce the NPE.
>
> * `` What is the difference...``
>
> The reviewer is correct in identifying the similarity between Tables 3 and 6. This was an oversight on our part. We apologize for any confusion this may have caused. Table 6 has been removed from the appendix in the revised version.

---

> > ### Comment · Reviewer_kwRu · 2024-08-07
> >
> > I thank the authors for the comments. However, I still have concerns regarding central points: I think that their method can be derived significantly simpler, and I am confused about the novelty of some results.
> >
> > - `Simplification`
> >
> > I cannot find an answer to my claim that the ExFM loss can be derived significantly simpler in the general answer. Can the authors please address my question explicitly? Since the authors agree that the derivation can be greatly simplified, I would in addition appreciate a high-level overview over the simplified variant and a proposed structure for the revised paper (in case the authors have already updated the manuscript available, it is often possible to post an updated PDF under anonymous link; ask the area chair for guidance).
> >
> > - `Novelty of results`
> >
> > I remain confused about the similarity to existing work, so let us establish a common understanding.
> >
> > The authors comment "that this formula is somewhat analogous to ours, as other similar formulas have also can be found in literature. However, we respectfully disagree that these are consistent with our contribution." So the existing results are analogous but inconsistent?
> >
> > Starting from the list of contributions in the Introduction, can the authors for each theoretical contribution (i) give the central lines where this contribution is achieved and (ii) what the closest known result is, together with a list of novel contributions?
> >
> >
> > - Minor point: `Jacobian independence`
> >
> > Can the authors say why the assumption in l. 172 is necessary and what it means?

---

> ### Author Response · Authors · 2024-08-09
>
> We thank Reviewer for additional important questions and for the discussion. We will answer them one by one.
>
> * ``Simplification
> I cannot find an answer to my claim that the ExFM loss can be derived significantly simpler in the general answer. Can the authors please address my question explicitly? Since the authors agree that the derivation can be greatly simplified, I would in addition appreciate a high-level overview over the simplified variant and a proposed structure for the revised paper (in case the authors have already updated the manuscript available, it is often possible to post an updated PDF under anonymous link; ask the area chair for guidance).``
>
> If one interested only in heuristics and want to derive some formula or method
> only for the practical verification, then he could go down the following path.
> Let us take the Eq. (8) from Lipman et al. "Flow Matching for Generative Modeling".
> Formally, we may substitute $p_t(x|x_1)$ from Eq. (10) and $u_t(x|x_1)$ from Eq. (15)
> from the same paper to this expression
> to get a special case of our formula.
> This is a straightforward way to derive our algorithm in the case of a Gaussian initial distribution.
> Note that this is highly intuitive reasoning.
>
> In contrast,
> in our work we used the following strict reasoning (somewhere implicitly).
> Our conclusions are based on
>
> a) Theorems from the cited Lipman et al on the equivalence of CFM and FM gradients (Theroems 1--2, Appendix A)
>
> b) theorems from Tong et al, "Improving and Generalizing Flow-Based Generative Models with Minibatch Optimal Transport"
>
> which extend the previous result to more general cases (Theorems 3.1--3.4, Table 1).
> Thus, these results allow us to start with the losses that are in CFM (in various modifications).
> Then we consider an _invertible map_ of type $\phi_{t,x_1}(x_0)=(1-t)x_0+tx_1+\sigma_stx_0$ and in order to proceed to the case of a simple _non-invertible_ map $\phi_{t,x_1}(x_0)=(1-t)x_0+tx_1$ we strictly consider the limit $\sigma_s\to0$. Thus, we maintain the rigor of the reasoning of a theoretical paper.
>
> Since there are a sufficient number of possible modifications of CFM (listed in Table 1 of Tong et al. or in Table 1 of our paper), and there are also many special cases, such as specific initial and target distributions as well as a stochastic modification of the equation in FM that is consistent with DDPM, we could not insert all these combinations in our paper.
> Rather, we have described in some detail a method that can be used to further obtain similar results to those we have already given.
> But to show the importance of the method, we have given some corollaries of the loss representation in a new form. For example a) a rigorous proof of variance reduction b) the comparability of numerical results of our algorithm, which directly follows from such a loss representation and the results of OT-CFM, which is sharpened for variance reduction c) an explicit record of trajectories for cases of Gaussian distributions (Eq. (36)), explicit recording of velocity for the case of Gaussian Mixture (Eq. (37)), a record of the score in integral form (Eq. (46)), the possibility of applying this technique to cases where both distributions are unknown and there are only samples from both distributions (Eq. (51)--(52)), etc.
> All of these results were obtained in one way or another from various loss modifications, similar to the one in native CFM, using our method described in detail in the main body of the paper.
>
> Because we hope that these results do not exhaust all possible applications of our technique, and because of page limitations, we have moved the specific results to Appendix, focusing in the main body of the paper on the method of obtaining them.
>
> Thus, in the main text we have detailed calculations so that it is clear how we get the result step by step.
>
> In the modified version of the article we added additional explanations that we discussed with other reviewers, tables, some figures we discussed in global answer and changed notations. The structure and the simplification of the article has not changed dramatically. We maintain all the nuances of the formulas' derivations as all these steps are the part of the strict analysis, which is also the aim of the paper. Revised version of the paper can be find in this anonymous link https://drive.google.com/file/d/17HDXtwWB505revPOXisi1F2lprECwUUu/view?usp=sharing.

---

> ### Author Response · Authors · 2024-08-09
>
> * ``Novelty of results
> I remain confused about the similarity to existing work, so let us establish a common understanding.
> The authors comment "that this formula is somewhat analogous to ours, as other similar formulas have also can be found in literature. However, we respectfully disagree that these are consistent with our contribution." So the existing results are analogous but inconsistent?
> Starting from the list of contributions in the Introduction, can the authors for each theoretical contribution (i) give the central lines where this contribution is achieved and (ii) what the closest known result is, together with a list of novel contributions?``
>
> Let us go through our contributions.
>
> 1. ``A tractable form of the FM loss is presented, which reaches...., but has a smaller variance``
> To the best of our knowledge, the result in the form of loss, which could reach zero and which would contain integral expressions from distribution functions, has not been found in the literature (unlike the expression for the vector field, see the next point below). So, apparently, the closest result that corresponds to this contribution is the already mentioned paper by Lipman et al, Eq. (6), about which the authors also write "Flow Matching is a simple and attractive objective, but na¨ıvely on its own, it is intractable to use in practice since we have no prior knowledge for what an appropriate $p_t$ and $u_t$ are."
> In addition, the rigorous analysis of variance that is mentioned in this contribution has not been encountered either.
>
> 2. ``The explicit expression in integral form for the vector field delivering the minimum to this
> loss (therefore for Flow Matching loss) is presented.``
> Partial cases, or analogs our Eq. (16), have been in various sources. The closest to ours is probably Eq. (4) and the unnumbered formula from Sec. 5.1 Liu et al, "Flow Straight and Fast: Learning to Generate and Transfer Data with Rectified Flow.".
> Also a formula similar to our Eq. (17) with softmax function is found in Scarvelis et al. "Closed-form Diffusion Models", Eq. (2) and others.
> But we note that in all the works in which similar formulas are found a) this formulas are not strictly derived b) was given one specific formula; on the contrast, we have proposed a whole family of formulas for the vector field (and score), starting from the case where we evaluate the integral not with SIS but with rejection sampling (Appendix B) and ending with formulas for DDPM-like models, i.e., for ODEs containing a stochastic term, namely the expression for the vector field in Eq (44) or the expression for the score (46). We have not encountered such expressions in the literature.
>
> 3. ``As a consequence, we derive expressions for the flow matching vector field and score in several particular cases (when linear conditional mapping is used, normal distribution, etc.);``
> We refer here, for example, to the results presented in Fig. 2, Figs. 4--6 (together with the analytical expressions by which they were obtained). We have not encountered similar formulas, similar studies on, for example, Gaussian Mixure separation we cited in the introduction ([15, 8]), but these studies are far from ours in terms of ideology.
>
> 4. ``Analytical analysis of SGD convergence showed that our formula have better training variance on several cases``.
> In this case, we are talking about Theorems ans resultd from Sec. 2.4 "Irreducible dispersion of gradient for CFM optimization". To the best of our knowledge, no similar studies have been conducted.
>
> 5. ``Numerical experiments show that we can achieve better learning results in fewer steps``.
> The topic of variance reduction in Flow-Matching-like models is raised most closely in the series of papers by Tong et al. on OT-CFM, where this heuristic technique was created specifically for practical implementation. We are compared with this technique in experiments in different dimensions. It turns out that in many cases we are not inferior to the results of this heuristic. However, our algorithm was rather a side effect of our theoretical calculations. It may well be accelerated if we use more efficient techniques (besides SIS or rejection sampling) for evaluating the integral.

---

> ### Author Response · Authors · 2024-08-09
>
> * ``Minor point: Jacobian independence
> Can the authors say why the assumption in l. 172 is necessary and what it means?``
>
> This condition is non-critical, it just simplifies the formula for the vector field. Consider the expression for the probability density function of the intermediate point L97 (in the old notations):
>
> $\rho_{x_1}(x,t)=[\phi_{t,x_1}]_*\rho_0(x):=\rho_0\big(\phi_{t,x_1}^{-1}(x)\big)\det\big[ \partial\phi_{t,x_1}^{-1}(x)\big/\partial x\big].$
>
> When we substitute this expression into the conditional distribution density function (7), the Jacobian under consideration will be in both the numerator and the denominator. But in case it does not depend on $x_1$, in the denominator it can be taken out of the integral and it will reduce with the same expression from the numerator.
> Since all important maps under consideration have this property, we considered mainly such simplified formulas (without explicitly writing out the Jacobian).
> In the case of the general form of the Jacobian, we came out to the expressions like Eq. (33)--(34), where the Jacobian enters explicitly.

---

> > ### Comment · Reviewer_kwRu · 2024-08-13
> >
> > ### Simplification
> >
> > I read the answer as follows: The authors do not agree that the derivation can be simplified, and stick with the existing derivation although. This is in contrast to their earlier answers suggesting that they indeed found a simpler derivation:
> >
> > > This algorithm itself could have been obtained in a simpler way.
> >
> > > We agree that the loss itself could be obtained more easily and with less computation. We have adressed this comment in the global answer.
> >
> > I would advise the authors to adopt a more direct communication style in the future.
> >
> > Since I did not come up with an alternative derivation, I consider this point as addressed but strongly advise the authors to try and simplify the derivation for an improved version of the article. I think this will significantly improve the readability and chances of adoption. In my view, a promising starting point would be via identities of expectation values.
> >
> > ### Contributions
> >
> > Thanks for this list of contributions. I agree that these results are in the paper and highlight that the two-batch-size algorithm is a useful training strategy, as evidenced by the reduced variance. I also think that the derived closed-form expressions are useful.
> >
> > ### Conclusion
> >
> > Overall, the paper makes a good case for two-batch-size training and has some interesting analytical results. However, the presentation also in the updated version is hard to follow. I therefore increase my score only slightly, to a borderline accept.

---

> > > ### Author Response · Authors · 2024-08-14
> > >
> > > We thank the reviewer for the time and effort. We are grateful for the increased score!
> > >
> > > We will take the Reviewer’s comments into account as we work to enhance the overall clarity of the presentation. We aimed to maintain a high level of rigour in the derivations to align with the theoretical nature of the work. We understand that this might have hindered readability in some parts. In the revised version, we are going to incorporate intuitive, non-rigorous explanations (simplified derivations of the formulas) to provide a clearer overview before diving into the mathematical details.

---

### Official Review · Reviewer_LEat · 2024-07-10

**Soundness:** 2
**Presentation:** 2
**Contribution:** 1
**Rating:** 2
**Confidence:** 5

**Summary:**

The paper proposes an analytic formula for the vector field satisfying the continuity equation for the given change of density which interpolates between two distributions in the sample space. This is a common setting in the Flow Matching model [6] (Rectified Flows [1], Stochastic Interpolants).

The authors apply the formula for several special cases like linear interpolation between samples and propose to use this formula for the training. They study the proposed training procedure empirically for synthetic examples and CIFAR-10 images.

[1] Liu, Xingchao, Chengyue Gong, and Qiang Liu. "Flow straight and fast: Learning to generate and transfer data with rectified flow." *arXiv preprint arXiv:2209.03003* (2022).

[6] Lipman, Yaron, Ricky TQ Chen, Heli Ben-Hamu, Maximilian Nickel, and Matt Le. "Flow matching for generative modeling." *arXiv preprint arXiv:2210.02747* (2022).

**Strengths:**

The presentation of the proposed method is clear.

**Weaknesses:**

The proposed formula is already well-known in the community. It was proposed in [1] (see eq. 4, Def 3.1, Sec 5.1 for demonstration) and [2] (see Appendix D). In [3] (see eq. 4), the authors propose to use an analogous formula for training a generative model and conduct a much more exhaustive empirical study. Moreover, [4,5] develop different applied methods building upon this formula.

The downside of this formula is also very well-known in the community, i.e. one has to use large batch sizes to accurately estimate the vector field, which comes with a significant computational cost. Moreover, for any finite size, the estimation of the vector field is biased and the estimation of the loss is biased which would deteriorate generation quality when used to train large-scale models. This limitation is not adequately studied in the paper, e.g. the models are not compared in terms of the training time and memory used.

Given that the main contribution of the paper has already been proposed and the empirical study of this formula is unsatisfactory, I cannot recommend this paper for acceptance.

Minor comments:

- The authors refer to the original FM framework [6] as Conditional Flow Matching, which was proposed in [7].
- The objective proposed in Flow Matching is already tractable, so it is not clear what “a tractable form of the Flow Matching objective” means.
- There is a typo in line 65.
- The derivative notation used is inconsistent with its description in the text.
- From eq. 19, I assume that “dispersion” means “variance”.

[1] Liu, Xingchao, Chengyue Gong, and Qiang Liu. "Flow straight and fast: Learning to generate and transfer data with rectified flow." *arXiv preprint arXiv:2209.03003* (2022).

[2] Neklyudov, Kirill, Rob Brekelmans, Daniel Severo, and Alireza Makhzani. "Action matching: Learning stochastic dynamics from samples." In *International conference on machine learning*, pp. 25858-25889. PMLR, 2023.

[3] Xu, Yilun, Ziming Liu, Max Tegmark, and Tommi Jaakkola. "Poisson flow generative models." *Advances in Neural Information Processing Systems* 35 (2022): 16782-16795.

[4] Scarvelis, Christopher, Haitz Sáez de Ocáriz Borde, and Justin Solomon. "Closed-form diffusion models." *arXiv preprint arXiv:2310.12395* (2023).

[5] Xie, Tianyu, Yu Zhu, Longlin Yu, Tong Yang, Ziheng Cheng, Shiyue Zhang, Xiangyu Zhang, and Cheng Zhang. "Reflected Flow Matching." *arXiv preprint arXiv:2405.16577* (2024).

[6] Lipman, Yaron, Ricky TQ Chen, Heli Ben-Hamu, Maximilian Nickel, and Matt Le. "Flow matching for generative modeling." *arXiv preprint arXiv:2210.02747* (2022).

[7] Tong, Alexander, Nikolay Malkin, Guillaume Huguet, Yanlei Zhang, Jarrid Rector-Brooks, Kilian Fatras, Guy Wolf, and Yoshua Bengio. "Improving and generalizing flow-based generative models with minibatch optimal transport." *arXiv preprint arXiv:2302.00482* (2023).

**Questions:**

I have no questions for the authors.

**Limitations:**

The authors do not provide a necessary literature review nor study the limitations of the proposed approach (see Weaknesses section above).

---

> ### Author Rebuttal · Authors · 2024-08-07
>
> We thank the reviewer for the specific notes. We will be happy to answer any additional
> questions you may have.
> **Let us discuss your comments:**
>
> * `` ... formula is already well-known...``
>
> Thank you for the links and the comment.
> We want to emphasize that we do not present formula (10) (or (11)) as our new main result, but a whole family of formulas, some of which are presented in Table 1, in particular, the situation with an additional stochastic term (Eq. (44) for vector field as well as Eq. (46) for the score).
>
> Let us describe in detail the difference between our work and the cited ones.
>
> We cited paper [1] as [10]. Eq. 4 there is a special case of our formula for linear map.
> Def. 3.1 is common definition, one needs to know the explicit form of $X(t)$ to get an explicit formula for the velocity.
> Sec. 5.1 describes only (biased) self-normalized importance sampling. We (in Appendix B) propose to use unbiased rejection sampling as well.
>
> Appendix D in the paper [2]  Neklyudov et al.
> states
> "In this section, we find velocity ... the conditional $k_t(x_t\mid x)$ is a Gaussian distribution"
> In contrast, we consider _any_ initial and target distributions as well as arbitrary conditional.
>
> We respectfully disagree, that Eq.3 in [3]  Xu et. al is similar to ours. It is completely different from our result.
> Moreover, in this paper
> the Poisson equation (Eq. 1 in [3]) is solved in an extended (of dimension $d+1$) space.
> This is fundamentally different from the usual flow equation studied in many Flow Matching articles.
> In addition, we focused on theoretical research, in particular, we rigorously argue for less variance in our approach (we submitted to ``Primary Area: Learning theory``).  Thus, conducting extensive experiments is not the goal of our paper.
>
> The paper [4] uses a different loss, and, in addition, it does not analyze the implications of the explicit formula, such as the analysis of variance, explicit expressions for the vector field, etc.
>
> Paper [5] was published on arXiv on May, 26  2024, this is later than we sent our full paper to NIPS-2024 (May, 22).
>
> * `` The downside of this formula... use large batch sizes...``
>
> One of the main contributions of our paper from a practical point of view is the use of two batch sizes. One batch size is used in training, it defines the number of members in a loss. It determines how much memory will be used when computing gradients in backpropagation.This size of the batches we have roughly coincides with the one used in similar experiments (such that the the amount of memory used is about the same).The second batch size determines how the right-hand side of the formula for the vector field (e.g., Eq. (15)) will be computed.Since no neural networks is involved in this calculation, we can take the batch size much larger than the first one. Even if there is not enough memory to get this sum at once,its computation can be easily divided into successive computations as well as parallelized.In our numerical experiments, the size of the second batches was significantly ($\sim10^1$--$10^2$ times) larger than the first one.The difference in training time depends on the model used. The time spent on backpropagation is the same if the first batch size is the same. In our algorithm there is additional time for right-hand side computation, but usually this addition is insignificant in case of "heavy" models.
>
> We believe a more in-depth analysis of the computational aspects is beyond the scope of this paper but is certainly a valuable direction for future work.
>
> * `` The authors refer...``
>
> Thank you for pointing this out. We correct the corresponding part of the paper in the revised version.
> We would like to note that, to the best of our knowledge, the original authors of [6] did not publicly release code. Consequently, we opted to utilize the implementation from [7] as a solid foundation for our research, as it closely adheres to the original framework.
>
> * `` ...not clear what “a tractable form of the Flow Matching objective” means....``
>
> By "tractable" we meant to write the loss in a way that immediately allows one to write the discrete loss as Eq. (13) or using other techniques from Appendix B to estimate the integral.For example, the possibility of using a different batched size to estimate this integral than the size when training the model. To the best of our knowledge, such a training scheme has not been described in the literature before. In addition, our notation of loss allows, for example, one to obtain explicitly an expression for the vector field (Eq. (37)) in the case of a Gaussian initial distribution and Gaussian Mixture as the target distribution. The study of the Gaussian separation problem is still important in the field of Flow Matching and Diffusion Models and, to the best of our knowledge, such an explicit expression for the velocity has not yet been described.
>
> We have added a corresponding explanation in the modified version of the article.
>
> * `` The derivative notation...``
>
> On L72 we define "Hereinafter the dash indicates the time derivative".
> In the vast majority of cases,
> dash stood at the expression of $\phi^\prime_{t,x_1} (x0)$ and denoted the derivative of the time parameter $t$.
> In Eq. (13) dash means derivative with respect to spatial variable $t_j$.
> There are no ambiguities.
> In other places, as p24, L624 or L767, we define dash explicitly.
>
> * `` The authors do not...``
>
> The introduction provides a solid foundation and links to relevant literature, including works with comprehensive overviews such as Tong et al. (2023b). However, it's important to note the relative scarcity of existing methods tailored to this specific type of analysis. To this end, we have provided a theoretical analysis of potential practical limitations in Appendix B. Most of our theoretical results are formulated in the form of Theorems or Propositions, thus, their formulation contains the conditions under which they are true.

---

> > ### Comment · Reviewer_LEat · 2024-08-12
> > **rebuttal acknowledgment**
> >
> > Thank you for your response. My main concerns remain unaddressed. The main proposed formula is very well known and such minor contributions as using two different batch sizes for estimating two different expectations must be followed up by an extensive empirical study. As a minor comment, I suggest the authors check on the definition of dash https://en.wikipedia.org/wiki/Dash.

---

> ### Author Response · Authors · 2024-08-12
>
> We thank the reviewer for pointing out weaknesses in our paper. At the same time, we respectfully disagree that our contribution consists only of an alogorithm with two different batch sizes. We listed the contribution of the paper in the first bullet in our general reply to all reviewers, and noted that it is much more than a single (already known) formula or a specific algorithm given.
> We tried to address all of your concerns in our rebuttal attempt. So, if you specify which of our answers were unconvincing, it would be very helpful for us.
>
>
> We also apologize for misunderstanding with notation. The point is that we made a mistake with the word ``dash`` (not with the essence of the notations, as we thought at first). To avoid ambiguity, we replaced the corresponding text with the following "Hereinafter the symbol "${}^\prime$" indicates the time derivative:...".

---

### Official Review · Reviewer_ctfF · 2024-07-15

**Soundness:** 1
**Presentation:** 1
**Contribution:** 2
**Rating:** 3
**Confidence:** 3

**Summary:**

The paper proposes a novel approach to training flow-based generative model by deriving the conditional flow matching objective function with respect to the flow function. The author argues that this new method of training will reduce variance, add stability, and ultimately lead to faster convergence. Additionally, the reformulation allow derivation of the exact vector field expression, and in some simple cases, enables the computation of the oracle trajectory solution.

**Strengths:**

- If the derivations are correct, this methodology could potentially add some innovations in the field of flow matching.

**Weaknesses:**

- The paper is difficult to follow and lacks clear writing and organization. Specifically, the authors' use of notations is very confusing.
- The mathematical computations do not appear to be very rigorous and some assumptions seem very incorrect. I may have misunderstood some derivations, so please correct me if I'm wrong (see Questions section).
- The experimental results are not very robust or convincing. For instance, while the paper proposes that one of their contributions is the reduction of variance during training, many of the results demonstrate larger variance across numerous, different metrics.
- Overall, the paper feels like it requires substantial revisions and is far from being polished.
Typos:
- Line 65, rho_1$\rightarrow$ $\rho_1$.
- Line 130, practical $\rightarrow$ practical form.

**Questions:**

- I find Eqn. 2 confusing. The authors should either replace the second term with the conditional vector field notation or explicitly state that the second term represents a derivative with respect to $t$.
- Eqn. 4 does not seem correct. Why can you rewrite the density function $\rho_j(x_1, x_t, t) = \rho_{x_1}(x_t, t)\rho_1(x_1)$? The distribution of the random variable $X_t$ is dependent on the distribution of $X_1$, so how can we write these terms as two independent variables.
- Also, in Eqn. 4 I believe $\rho_{x_1}(x_t, t)\rightarrow\rho_{t}(x_t, t)$. What does subscript $x_1$ mean? Does this mean it is some conditional distribution? This notation seems ambiguous.
- Why do you use the notation $p_m(\cdot)$ and not $p_t(\cdot)$. The distribution is different with respect to $t$, using $m$ makes it seems like a joint distribution of all $t$ values.
- Have the authors tried running experiment on non-Gaussian priors?
- Rather than explaining the training scheme in detail, I would suggest adding pseudo code.

**Limitations:**

The author has adequately addressed the limitations.

---

> ### Author Rebuttal · Authors · 2024-08-07
>
> We thank the reviewer for their detailed review and comments.
>
> **Let us also discuss your comments:**
>
> * ``The paper is difficult to follow...``
>
> We have carefully revised the manuscript to enhance its readability and coherence. We make several changes in the notation which we put in one-page PDF in the general answer. While striving for balance in the theoretical analysis, we have included all necessary proofs to support our findings. Regarding notations, we have employed mostly standard probability notations from of mathematical analysis and probability theory.
>
> * ``The experimental results... ``
>
> While we highlight variance reduction as a key contribution, we understand the limitations of standard metrics like W2, NLL, and Energy distance. To address this, we have included supplementary visualizations that provide a more intuitive understanding of sample quality. We are eager to explore additional metrics suggested by the reviewer to further strengthen our findings.
>
> Our loss function consistently exhibits lower values across most datasets, indicating reduced variance compared to baselines. However, we admit that there are instances, particularly in some tabular datasets (especially BSDS dataset), where this reduction is less pronounced.
> To provide a more comprehensive understanding of our method's behavior, the paper includes additional figures illustrating the evolution of loss and metrics over training steps (Figures 7, 9, 10, and 8, 11, respectively). These visualizations offer compelling evidence of the variance reduction achieved during training.
>
> We have incorporated additional analyses for toy 2D data, including Energy Distance and loss over steps, as well as expanded visual comparisons and datasets, which also includes density comparisons of distributions. These results are presented in a revised version of the paper and the results of expanded visual comparison can be already viewed in the new supplementary 1-page PDF. Our experiments clearly indicate that our method consistently outperforms CFM and, in most cases, OT-CFM, which was specifically developed to address variance.
>
> **Answering your questions:**
> * ``I find Eqn. 2 confusing...``
>
> On the L72 we write: "Hereinafter the dash indicates the time derivative.". We added additional mathematical explanation of the dash symbol to the revised version of the paper : $\phi^\prime_{t,\cdot} (X):=\frac{d}{dt}\phi_{t,\cdot} (x)|_{x=X}$.
>
> * ``Eqn. 4 does not seem correct... ``
>
> The density $\rho_{x_1}$ is defined in L97:
> $\rho_{x_1}(x,t)=[\phi_{t,x_1}]_{*}\rho_0(x)$.
>
> The dependence of the value $x_1$ of the random variable $X_1$ is emphasized by the lower index $x_1$.
> Indeed, different densities have the same letter for designation ($\rho$) in our paper, and the meaning is clear only from the indices.
> We tried to avoid notations like $A|B$ as much as possible,
> since such notations are used for events or random variables,
> while we are narrating in the stricter terms of integrals,
> so the arguments of the functions $\rho$ have the meaning of specific values of random variables,
> not the random variables itself.
>
> * ``Also, in Eqn. 4...``
>
> As stated in the answer to the previous question, in this case $x_1$ in the index has the meaning of the dependence of the density function $\rho_{x_1}(\cdot, t)$ on the value $x_1$ of the random variable $X_1$.
> However, we have changed the notations to more convenient ones.
>
> * ``Why do you use...``
>
> We change the notations and listed changes in the 1-page PDF in the "all reviewers answer".
> From these new notations one can clearly see what we consider to be the argument of the function and what is  parameter(s).
>
> Since we considered $\rho$ and similar functions as functions from the point of view of mathematical analysis and worked more with integrals than with mathematical expectations, from this point of view it does not matter where to write the parameter -- in the lower index, or as an argument; the choice of one or the other designation is dictated by convenience.
>
> We consider probability density function $\rho_m(x_t,t)$ as a function of the first argument,
> considering $t$ as a parameter.
> In Flow Matching theory, the time variable is also may be treated as a random variable, so the interpretation of the $\rho_m(x_t,t)$ function as a joint density distribution is legitimate.
> However, in our calculations, we have only the integral over time as an external integral and do not give the variable $t$ the meaning of a random variable (unlike, of course, the random variables corresponding to the initial and target distributions).
>
> * ``Have the authors tried... ``
>
> Yes, Table 4 summarizes the results for the case when neither initial nor final distributions are known,
> but only point clouds are given.
> The moons type was considered as the initial distribution in some of this experiments.
> In this case, we work not with the original formula for the vector field, but with its modification Eq. (51).
>
> For the original version of the discrete loss Eq. (14) with vector field given as Eq. (16),
> we tried other distributions as $\rho_0$, in particular, the uniform distribution.
> But such replacement did not give any advantages, moreover, the use of Gaussian distribution allows us to use rather fast and accurate SoftMax function realisation.
> Thus, we did not investigate in this direction and give no numerical results in the paper.
>
> * ``Rather than explaining...``
>
> Thank you for the suggestion.
> We believe the detailed algorithms in Appendix C offer a comprehensive understanding of the training process.
> But it did not fit into the main text along with its description due to space limitations. Since the paper is focused on theoretical calculations, we decided to remove the practical algorithm to the Appendix in this version of the article.
> We will return it to the main text if it is possible to use an additional page.

---

> > ### Comment · Reviewer_ctfF · 2024-08-13
> >
> > Thank you for addressing the notation issue. While the improvement is noted, I believe the paper could benefit from further refinement in terms of overall organization, writing clarity, and consistent notation. As such, I will be maintaining my current score.

---

> > > ### Author Response · Authors · 2024-08-14
> > >
> > > We thank the reviewer. We acknowledge the areas identified for potential improvement and will carefully consider these points for future revisions.

---

### Official Review · Reviewer_xno6 · 2024-07-17

**Soundness:** 3
**Presentation:** 1
**Contribution:** 2
**Rating:** 3
**Confidence:** 4

**Summary:**

This works aims at producing a lower variance loss for flow matching. This is done by using the formula for the ground truth marginal velocity field, estimating it using self-normalized importance sampling, then regressing onto this estimated marginal velocity field.

**Strengths:**

Variance reduction for CFM is a useful avenue of research

**Weaknesses:**

- This paper lacks polish. I felt this paper is cumbersome to read, with a lot of heavy notation that can be drastically simplified, whereas the proposed algorithm is quite simple.
- The proposed approach is not exactly novel. The "Explicit Flow Matching" objective is just the original "Flow Matching" objective where we regress onto the optimal velocity field. The practical implementation being proposed is a simple application of self-normalized importance sampling.
- Importantly, the proposed approach leads to a biased objective (when estimated through a minibatch) where the optimum is not guaranteed to be the correct velocity field. This is not a problem for the low-dimensional experiments but importance sampling becomes more problematic in high dimensions.

**Questions:**

The ExFM objective (Eq 8 in this paper) is the same as the FM objective [Ref. 1, Eq 5] where u_t is given by the marginal velocity field [Ref. 1, Eq 8], equivalent to Eq 10 in this paper. That this FM objective has the same gradient as CFM (Theorem 2.1 in this paper) is also stated in [Ref. 1, Theorem 2].

Given this, I don't think ExFM should be treated as a new loss.

The use of self-normalized importance sampling (SIS) for estimating the marginal velocity field could be interesting in its own right. However, it is misleading to state that this objective can reach zero because of the need for estimating the marginal velocity field. SIS introduces bias in favor of reducing variance. There should be some analysis on the bias introduced by the objective, comparing the learned model to the marginal velocity field.

[1] "Flow matching for generative modeling"

**Limitations:**

The proposed ExFM objective is equivalent to the original FM objective, and the paper should not over-claim this contribution. This objective is intractable, so the use of smart estimation methods is interesting in its own right.

---

> ### Author Rebuttal · Authors · 2024-08-07
>
> Dear reviewer, thank you very much for your analysis of our work and the specific notes you
> formulated! If you have any additional questions, we will be happy to answer them and make
> additional edits to the text of the manuscript to improve the quality of the presentation.
>
> **Let us discuss your comments:**
>
> * `` This paper lacks polish... ``
>
> We thank the reviewer for their valuable feedback. We agree that clarity is essential, and we have carefully revised the text to enhance readability.
> We make several changes in the notation which we put in 1-page PDF document in the general answer.
> While the proposed algorithm is indeed straightforward, the core contributions of this paper lie in the theoretical analysis of Flow Matching.
> This analysis necessitates a certain level of mathematical notation to ensure rigor and precision. We have made every effort to simplify notation where possible without compromising the clarity of the proofs. We believe that a rigorous analysis is essential for the broader community to understand the algorithm's implications and potential extensions.
> We emphasize, as we write about it in Introduction, that the new loss we obtained is only a consequence of our analysis and not the main result. Of course, this loss itself could have been obtained more quickly, but the point of our study is to develop a method that allows one, for example, to obtain the various modifications of the exact formulas collected in Table 1.
>
> * `` The proposed approach is not exactly novel... ``
>
> We thank the reviewer for their insightful comments. We agree that our proposed loss function can be seen as a special case of the Flow Matching objective.
> Analogs of our formula are given in other papers and we have cited these papers.
>
> However, to the best of our knowledge, the points listed in the general response to the reviewers are new and represent our main contributions.
> Thus, our investigation goes far beyond presenting a single formula for a vector field like Eq (15) or a single practical algorithm.
>
> *  `` Importantly, the proposed approach leads to a biased objective...``
>
> We appreciate the reviewer's concern about the potential bias of our objective function in high-dimensional settings.
> Indeed, SIS may possess bias. To circumvent the issue, in addition to using SIS directly, we use rejection sampling, which removes the denominator in the velocity formula, see Appendix B ''Estimation of integrals''.
> Thus, our variance reduction method has a non-bias modification as well.
> Note that no bias was observed in the experiments using SIS, so we performed most of the experiments using this technique, as it was computationally more efficient.
> We have added an explicit mention of rejection sampling in the main text to avoid misunderstandings.
>
> In addition to this, as correctly pointed out, flow matching approaches, including extensions based on Optimal Transport (OT), generally suffer from challenges in high dimensions.
> While we acknowledge the limitations of our current approach in handling extremely high-dimensional data and long time horizons, we believe that our work provides valuable insights into the theoretical underpinnings of the problem. We have conducted a thorough analysis of the bias in our objective, as detailed in Section 2.4, and demonstrated the effectiveness of our method in lower-dimensional settings. We consider extending our approach to higher dimensions and longer time horizons as promising future work.
>
> **Answering your questions:**
>
> * ``The ExFM objective is the same as the FM objective ... ``
>
> We agree that there are similarities between ExFM and FM objectives. But as author in [1] (Lipman et al. 2022) said in the discussion of this loss ``Flow Matching is a simple and attractive objective, but naїvely on its own, it is intractable to use in practice since we have no prior knowledge for what an appropriate $p_t$ and $u_t$ are.''
> Our work provides an explicit, integral form of the FM loss, enabling direct algorithm development based on the discrete loss Eq. (14).
> Moreover, Theorem 2.1 and 2.5 offer novel insights into the relationship between ExFM and CFM, demonstrating its potential advantages such as smaller variance.
>
> * ``The use of self-normalized importance sampling...``
>
> We agree that if we evaluate the integral over samples, this evaluation may not be accurate, and thus the discrete loss (for example, $L^d_{\hbox{\tiny ExFM}}$ in Eq. (14)) does not reach zero at the minimum. However, we were talking about reaching zero in the context of the exact value of the integral, i.e., in the case when we consider the loss like $L_{\hbox{\tiny ExFM}}$ in Eq. (8),
> where the expression through the integrals is subtracted from the expression for the model $v_\theta$ under the norm.
> In this case, at the exact value of the vector field, the integrals (which correspond to mat. expectations) yield zero in the final result.
>
> And as we stated above, in addition to SIS, we  proposed other, non-biased, ways of evaluating the integral in Appendix B.
> Some of our experiments have shown that using SIS gives no bias and performs even slightly better than our proposed rejection sampling based estimation. However, comparison of these methods is orthogonal to the main goals of our paper.
> Note that since our work has a theoretical focus (and we posted the paper to the area "Learning theory"), it was not our goal to find the best algorithm for approximate calculation of this integral. On the contrary, we believe that besides the SIS and unbiased rejections sampling we have proposed for the example, there are many other ways to improve the calculation of the integral in question (for example, the cited in our paper [3] Gabriel Cardoso et al. “BR-SNIS: Bias Reduced Self-Normalized Importance Sampling”).
> We hope that our proposed ideas will allow one to further modify existing algorithms or build new ones to work more efficiently.

---

### Author Rebuttal · Authors · 2024-08-07

We thank the reviewers for their valuable feedback and insights. To enhance the clarity and comprehensiveness of our work, we have identified key areas where multiple reviewers expressed similar questions or critiques. We will provide detailed explanations and refinements in the updated version of the paper.

   * We emphasize that our paper is theoretical (chosen area is Learning Theory), and the main results consist in the methodology of deriving various formulas for training a vector field, some of which are collected in Table 1 in the main paper, strictly proving the lower variance of the proposed loss, obtaining various modifications of the formula for training, including the case when neither the initial nor the final distributions are known, but only samples are given in the cases with stochastic addition in the equation. The obtained algorithm is rather a practical illustration of our theoretical studies than the main result. This algorithm itself could have been obtained in a simpler way. We also emphasize that the significance of the loss lies in its derivation from a rigorous analysis of flow matching, which enables us to develop a deeper understanding of the underlying dynamics and to explore various modifications (as exemplified in Table 1).

     The analysis goes beyond simply introducing a new loss function and includes a detailed examination of bias and variance, culminating in the derivation of an optimal velocity formula. We believe that this theoretical foundation is essential for advancing the field of continuous flows and informing the development of future algorithms.

     We agree that our proposed loss function can be seen as a special case of the Flow Matching objective.
      Analogues of our formula are given in other papers and we cited some of them.

     However, to the best of our knowledge,

        a) a learning algorithm that uses 2 different batches has not been published before;

        b) we propose to use not only SIS, but for example rejection sampling or bias-reduced versions of SIS ([3], Gabriel Cardoso et al.), as written in Appendix B;

        c) an explicit Flow Matching solution for the trajectories Eq (36) for the Gaussian->Gaussian case has not been obtained before;

        d) an explicit expression for the vector field Eq. (37) for the Gaussian->Gaussian Mixture case was not obtained before;

        e) there was no analysis of the trajectories in the case of an additional stochastic term as in Appendix E before;

        f) we consider a class of invertable conditional maps and consider rigour limit to get the results, valid for the non-invertable simple linear map ($x_0 (1-t) + x_1 t$);

        etc.

     Thus, our investigation goes far beyond presenting a single formula for a vector field like Eq (15) or a single practical algorithm.

   * While our paper is mostly focuses on the theoretical aspects, as a prove of concept we provide a more comprehensive evaluation, we have incorporated supplementary visualizations and expanded our analysis. Our variance reduction is less evident in certain tabular datasets, notably BSDS. To address this, we've included detailed visualizations of loss and metric evolution (Figures 7-11) which clearly demonstrate the variance reduction achieved during training. We have significantly enhanced the paper by incorporating additional analyses, visualizations, and datasets. This includes new experiments on 2D toy data using Energy Distance and loss over steps, as well as expanded visual comparisons (can be viewed in the supplementary 1-page PDF) and density comparisons of distributions. Our experimental results consistently demonstrate that our method surpasses the performance of CFM. Notably, it also matches or exceeds the performance of OT-CFM, a model explicitly tailored to variance reduction, in the majority of our evaluations.

   * We corrected inaccuracies in highlighting in the NLL metrics table, adjusted the format, but kept e-notation because we believe that maintaining e-notation offers a more compact and informative representation. The revised manuscript now includes a more in-depth analysis comparing CFM, OT-CFM, and ExFM using Wasserstein distance (Table 1 in 1-page PDF and Energy Distance (Table 2 in 1-page PDF).

   * We have significantly enhanced the paper with new analysis for toy 2D data. This includes plots illustrating the relationship between energy distance and steps, loss and steps, as well as expanded visual comparisons (Figure 1 in 1-page PDF) and comparison of distributions' densities. Here we provide the most important results of additional toy 2d data visualisations. Other mentioned additions are in the revised version of the paper. But from our experiments we can see the tendency that our  method outperforms CFM in all cases and in most cases OT-CFM, which was specifically built to reduce variance.

   * We sincerely apologize for any typos or misleading information present in the previous version of the paper. We have conducted a thorough review and made necessary corrections to ensure accuracy and clarity. We appreciate the reviewer's diligence in identifying these issues.

By addressing these points, we aim to demonstrate the value of our work in providing a strong theoretical foundation for flow matching, while also acknowledging the importance of clear communication and accurate presentation.

---

### Decision · Program_Chairs · 2024-09-25

**Decision:**

Reject

**Comment:**

This paper advocates for a novel loss (Explicit Flow Matching, ExFM) with the goal of reducing variance during training.

Below I summarize the main strengths and weaknesses of this paper. Overall, the reviewers find the goal of the paper useful, however the presented method is incremental, producing biased gradients, and the paper not sufficiently well written. For these reasons the paper cannot be accepted.


Strengths:
+ Reducing variance in FM training is a useful goal.
+ Importance sampling for FM loss.

Weaknesses:
- Method proposed is incremental: the loss is not new - it is the original FM objective and the target velocity is replaced with a mini-batch estimate.
- When velocity is estimated with mini-batch the method produce biased gradients, which can be a serious issue in high dimensions.
- Paper is hard to read and follow.